# A multi-scale cortical wiring space links cellular architecture and functional dynamics in the human brain

**Casey Paquola**[1]\*, **Jakob Seidlitz**[2], **Oualid Benkarim**[1], **Jessica Royer**[1], **Petr Klimes**[3], **Richard A. I. Bethlehem**[4], **Sara Larivière**[1], **Reinder Vos de Wael**[1], **Raul Rodríguez-Cruces**[1], **Jeffery A. Hall**[3], **Birgit Frauscher**[3], **Jonathan Smallwood**[5], **Boris C. Bernhardt**[1]\*

**1** Multimodal Imaging and Connectome Analysis Lab, McConnell Brain Imaging Centre, Montreal Neurological Institute and Hospital, McGill University, Montreal, Quebec, Canada, **2** Developmental Neurogenomics Unit, National Institute of Mental Health, Bethesda, Maryland, United States of America, **3** Montreal Neurological Institute and Hospital, McGill University, Montreal, Quebec, Canada, **4** Department of Psychiatry, University of Cambridge, Cambridge, United Kingdom, **5** York Neuroimaging Center, University of York, York, United Kingdom

\* casey.paquola@mail.mcgill.ca (CP); boris.bernhardt@mcgill.ca (BCB)

**Data Availability Statement:** The human connectome project dataset is openly available on https://db.humanconnectome.org/. The BrainSpace toolbox is available under https://github.com/

## Abstract

The vast net of fibres within and underneath the cortex is optimised to support the convergence of different levels of brain organisation. Here, we propose a novel coordinate system of the human cortex based on an advanced model of its connectivity. Our approach is inspired by seminal, but so far largely neglected models of cortico–cortical wiring established by postmortem anatomical studies and capitalises on cutting-edge in vivo neuroimaging and machine learning. The new model expands the currently prevailing diffusion magnetic resonance imaging (MRI) tractography approach by incorporation of additional features of cortical microstructure and cortico–cortical proximity. Studying several datasets and different parcellation schemes, we could show that our coordinate system robustly recapitulates established sensory-limbic and anterior–posterior dimensions of brain organisation. A series of validation experiments showed that the new wiring space reflects cortical microcircuit features (including pyramidal neuron depth and glial expression) and allowed for competitive simulations of functional connectivity and dynamics based on resting-state functional magnetic resonance imaging (rs-fMRI) and human intracranial electroencephalography (EEG) coherence. Our results advance our understanding of how cell-specific neurobiological gradients produce a hierarchical cortical wiring scheme that is concordant with increasing functional sophistication of human brain organisation. Our evaluations demonstrate the cortical wiring space bridges across scales of neural organisation and can be easily translated to single individuals.

## Introduction

Neuronal activity in the cortex is simultaneously constrained by both local columnar circuitry and large-scale networks [1, 2]. It is generally assumed that this interaction emerges because

MICA-MNI/BrainSpace. Preprocessed group-level matrices, normative manifold maps, and integral scripts are openly available at https://git.io/JTg1l; instructions on how to access the main datafiles and how to reproduce the figures is available in the readme file of that repository.

**Funding:** CP and RC were funded through a postdoctoral fellowship of the Fonds de la Recherche due Quebec – Santé (FRQ-S). OB was funded by a Healthy Brains for Healthy Lives (HBHL) postdoctoral fellowship. JR was supported by a Canadian Open Neuroscience Platform (CONP) fellowship. JS was supported by the European Research Council (WANDERINGMINDS-ERC646927). BB acknowledges research support from the National Science and Engineering Research Council of Canada (NSERC Discovery-1304413), the Canadian Institutes of Health Research (CIHR FDN-154298), SickKids Foundation (NI17-039), BrainCanada, Azrieli Center for Autism Research (ACAR-TACC), Helmholtz International BigBrain Analytics Learning Laboratory (HIBALL), FRQ-S, and the Tier-2 Canada Research Chairs program. The funders had no role in study design, data collection and analysis, decision to publish, or preparation of the manuscript.

**Competing interests:** The authors have declared that no competing interests exist.

**Abbreviations:** EEG, electroencephalography; EPI, echo-planar imaging; FDR, false discovery rate; GD, geodesic distance; HCP, Human Connectome Project; MICs, Microstructure Informed Connectomics; MNI, Montreal Neurological Institute; MPC, microstructure profile covariance; MRI, magnetic resonance imaging; OPC, oligodendrocyte progenitor cell; qT1, quantitative T1; rs-fMRI, resting-state functional magnetic resonance imaging; scTHS-seq, single-cell transposome hypersensitive-site sequencing; SIFT2, spherical-deconvolution informed filtering of tractograms; snDrop-seq, single-nucleus Droplet-based sequencing; T1w, T1 weighted; T2w, T2 weighted.

individual neurons are embedded in a global context through an intricate web of short- and long-range fibres. Developing a better model of this cortical wiring scheme is a key goal of systems neuroscience because it would serve as a blueprint for the mechanisms through which local influences on neural function impact on spatially distant sites and vice versa.

Postmortem histological and gene expression studies provide gold standard descriptions of how neurobiological and microstructural features are distributed across the cortex [3–6]. Histological and genetic properties of the brain often vary gradually together, mirroring certain processing hierarchies, such as the visual system [7]. This suggests that observed brain organisation may be the consequence of a set of consistent principles that are expressed across multiple scales (gene expression, cytoarchitecture, cortical wiring, and macroscale function). Critically, full appreciation of how local and global features of brain organisation constrain neural function requires that multiple levels of brain organisation are mapped in vivo. To achieve this goal, our study capitalised on state-of-the-art magnetic resonance imaging (MRI) methods and machine learning techniques to build a novel model of the human cortical wiring scheme. We tested whether this model provides a meaningful description of how structure shapes macroscale brain function and the information flow between different systems. In particular, if our model successfully bridges the gap between micro- and macroscopic scales of neural organisation, then it should describe local features of both cortical microcircuitry and its macroscale organisation and deliver meaningful predictions for brain function.

Currently, the prevailing technique to infer structural connectivity in living humans is diffusion MRI tractography [8, 9]. By approximating white matter fibre tracts in vivo [8–10], tractography has advanced our understanding of structural networks in health [11–13] and disease [14–16] and shaped our understanding of the constraining role of brain structure on function [17–22]. Diffusion MRI tractography, however, has recognised limitations [23, 24]. Crucially, the technique does not explicitly model intracortical axon collaterals and superficial white matter fibres, such short-range fibres contributing to >90% of all cortico–cortical connections [25]. To address this gap, our approach combines diffusion tractography with 2 complementary facets of cortical wiring, namely spatial proximity and microstructural similarity. Spatial proximity predicts short cortico–cortical fibres [26, 27], which facilitate the most common type of neural communication also referred to as "nearest-neighbour-or-next-door-but-one" [28]. Microstructural similarity is a powerful predictor of interregional connectivity in nonhuman animals [29], whereby the "structural model" of cortico–cortical connectivity postulates that connectivity likelihood between 2 regions is strongly related to similarity in cytoarchitecture [30–33]. We recently developed and histologically validated microstructure profile covariance (MPC) analysis, which quantifies microstructural similarity between different cortical areas in vivo through a systematic comparison of intracortical myelin-sensitive neuroimaging profiles [7]. These complementary features can be fused using manifold learning techniques, resulting in a more holistic, multi-scale representation of cortical wiring. This extends upon previous work, in which we and others have derived manifolds from single modalities to map gradual changes in functional connectivity or tissue microstructure [7, 34].

Here, we generate a new coordinate system of the human cortex that is governed by complementary in vivo features of cortical wiring, expanding on traditional diffusion MRI tractography. Our wiring space incorporates advanced neuroimaging measures of cortical microstructure similarity, proximity, and white matter fibres, fused by nonlinear dimensionality reduction techniques. We tested the neurobiological validity of our newly developed model by cross-referencing it against postmortem histology and RNA sequencing data [5, 35, 36]. Furthermore, we assessed the utility of our model to understand macroscale features of brain function and information flow by assessing how well the model predicts resting-state connectivity obtained from functional MRI as well as directed descriptions of neural function and

processing hierarchy provided by intracerebral stereo-electroencephalography. These experiments were complemented with a comprehensive battery of robustness and replication analysis to assess the consistency and generalizability of the new wiring model across analytical choices and datasets.

## Results

### A multi-scale model of cortical wiring

Cortical wiring was first derived from a "Discovery" subset of the Human Connectome Project (HCP) dataset ($n$ = 100 unrelated adults) that offers high-resolution structural MRI, diffusion MRI, and microstructurally sensitive T1 weighted (T1w)/T2 weighted (T2w) maps [37]. Main findings are presented on HCPs multimodal parcellation and replicated on a folding based parcellation (Fig 1A, S1 Fig; see Methods section for details).

The model integrated 3 complimentary features of structural connectivity, mapped between spatially contiguous nodes: (i) **Geodesic distance (GD)**, calculated as the shortest path between 2 nodes along the cortical midsurface, reflects the spatial proximity and cortico–cortical wiring cost of 2 regions [26]; (ii) **MPC**, which is the correlation between myelin-sensitive imaging profiles taken at each node in the direction of cortical columns [7], indexes architectonic similarity, the strongest predictor of projections in nonhuman primates [29]; and (iii) **Tractography strength (TS)**, based on tractography applied to diffusion-weighted MRI, yields an estimate of the white matter tracts between each pair of nodes. Regional variations in the correspondence of wiring features, in terms of both magnitude and direction, highlight the necessity of multifeature integration to provide a nuanced and expressive characterisation of a region's structural connectivity (S1 Fig).

To integrate these features into a compact coordinate system governed by cortical wiring, we normalised and concatenated the interregional matrices, computed an affinity matrix, and performed manifold learning (Fig 1B; see "Methods" section). Diffusion map embedding, a nonlinear dimensionality reduction technique, was selected as a fast and robust approach that provides a global characterisation while preserving local structure in a data-driven manner [38]. Two dominant eigenvectors explained approximately 61% of variance in wiring affinity, with the first illustrating an anterior–posterior gradient (approximately 36%) and the second a sensory-fugal gradient (approximately 25%). These gradients represent principle axes of variation in cortical wiring (Fig 1C). The 2D representation is hereafter referred to as the "wiring space." Distances between 2 nodes in this new space provide a single integrative metric of cortical wiring affinity (Fig 1D). Nodes with high wiring affinity are close by, whereas dissimilar regions have a greater wiring distance from other regions. Analysing the average wiring distance of each node to all other nodes (calculated as distance in the 2D manifold), we found that primary sensory areas, such as the calcarine sulcus and superior precentral gyrus, exhibited the most distinctive, specialised cortical wiring (Fig 1D).

To evaluate generalizability, we reconstructed the wiring model in an independent dataset of 40 healthy adults scanned at our imaging centre (Microstructure Informed Connectomics (MICs) cohort; see Methods section for details). While imaging parameters were comparable to the main cohort, this replication cohort involved acquisition of quantitative T1 relaxometry data to index intracortical microstructure instead of T1w/T2w maps [39–42]. Regardless of these site-wise idiosyncrasies, our procedure produced highly similar wiring spaces (correlations between both sites for eigenvectors 1/2: r = 0.81/0.81; S3A Fig). The wiring space was also conserved at an individual level (S4A Fig). The most prominent interindividual shifts in nodal positioning were observed in superior parietal and orbitofrontal regions (S4C Fig).

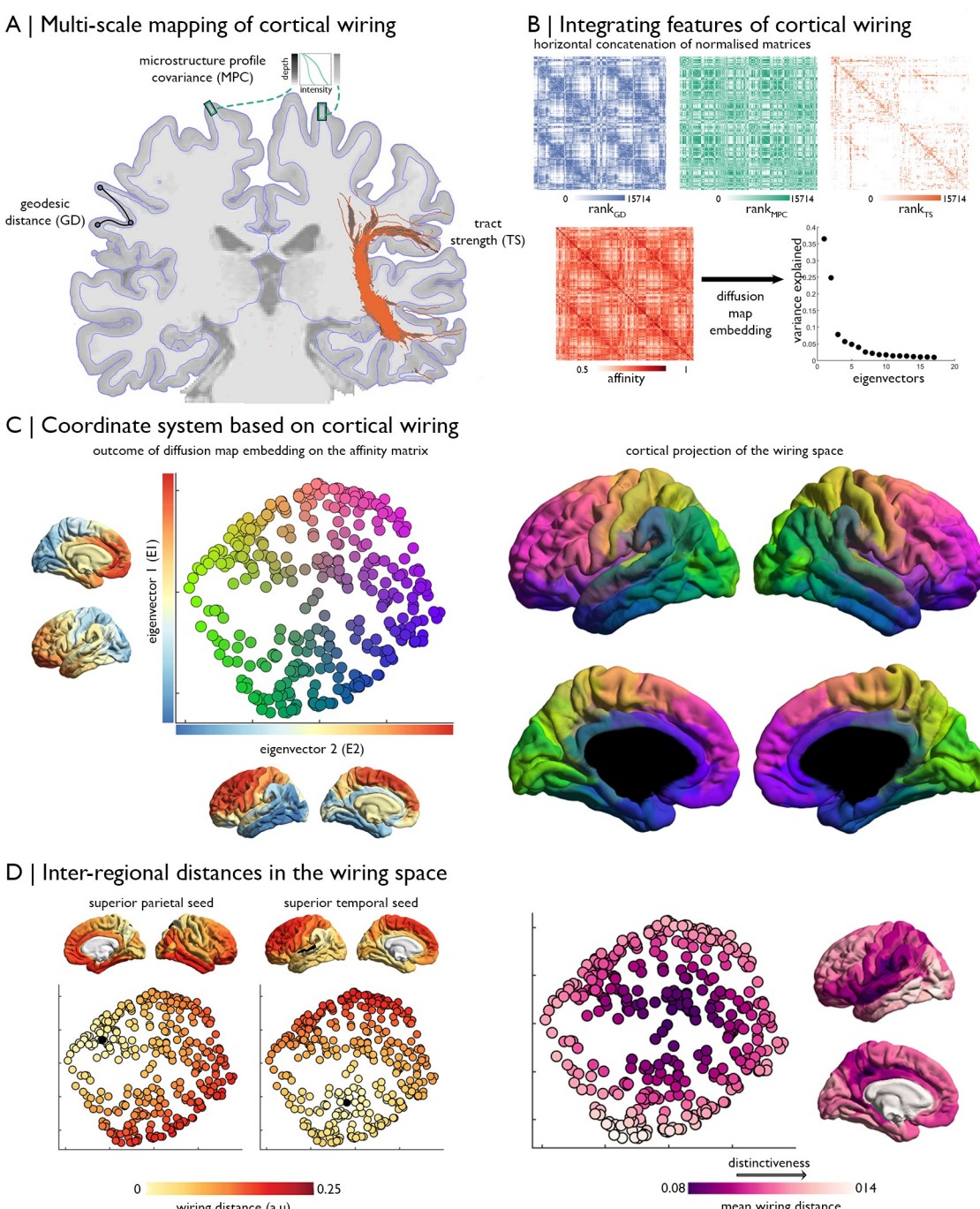

**Fig 1. The multi-scale cortical wiring model. (A)** Wiring features, i.e., GD, MPC, and diffusion-based TS were estimated between all pairs of nodes. **(B)** Normalised matrices were concatenated and transformed into an affinity matrix. Manifold learning identified a lower-dimensional space determined by cortical wiring. **(C)** Left: node positions in this newly discovered space, coloured according to proximity to axis limits. Closeness to the maximum of the first eigenvector is redness, towards the minimum of the second eigenvector is greenness, and towards the maximum of the second eigenvector is blueness. The first 2 eigenvectors are shown on the respective axes. Right: equivalent cortical surface representation. **(D)** Calculation of interregional distances in the wiring space from specific seeds to other regions of cortex. Overall distance to all other nodes can also be quantified to index centrality of different regions, with more distinctive areas having longer distances to nodes. Replication with the Freesurfer-style preprocessing pipeline in S1 Fig. Essential data are available on https://git.io/JTg1l. a.u., arbitrary units; E, eigenvector; GD, geodesic distance; MPC, microstructure profile covariance; TS, tractography strength.

## Neurobiological underpinnings

We next evaluated the capacity of the new model to reflect local neurobiological features by examining postmortem human histology and gene expression data. We generated cell-staining intensity profiles for each node from a high-resolution volumetric reconstruction of a single Merker-stained human brain [35] (Fig 2, S5 Fig) and extracted gene expression from mRNA sequencing data in 11 neocortical areas, each matched to 1 node [36, 43] (Fig 3, S6 Fig). Cytoarchitectural similarity and gene co-expression were correlated to wiring distance (histology: r = −0.40, $p < 0.001$ Fig 2B; co-expression: r = −0.36, $p < 0.001$).

We hypothesised that the principle axes of the cortical wiring scheme would describe systematic variations in cytoarchitecture that reflect a region's position in a neural hierarchy. It has previously been proposed that externopyramidisation is optimally suited to assess hierarchy-dependent cytoarchitecture because it tracks the laminar origin of projections, which signifies the predominance of feedback or feedforward processing [5]. Externopyramidisation was estimated from histological markers capturing the relative density and depth of pyramidal neurons [44] (Fig 2A), the primary source of interregional projections. Increasing values reflect a shift from more infragranular feedback connections to more supragranular feedforward connections [5, 44]. Multiple linear regression indicated that the eigenvectors of the wiring space explained substantial variance of externopyramidisation (r = 0.51, $p_{spin} < 0.001$; Fig 2C) and was independent of regional variations in cortical morphology (S7A Fig, $p_{spin} > 0.05$). Notably, the complete model that aggregated all wiring features (i.e., MPC, GD, TS) in a low-dimensional space explained more variance in externopyramidisation than models constructed using alternative combinations or subsets of the wiring features (Table A in S1 Text). Externopyramidisation gradually decreased along the second eigenvector, suggesting that our wiring space captures a posterior to anterior transition from feedforward to feedback processing.

We further explored how the broader cellular composition and microcircuitry relates to the layout of the wiring space. To do so, we estimated the expression of 8 canonical cell type gene sets in 11 cortical areas (see Methods section) and found that these expression patterns accounted for significant variance in the macroscale organisation of cortical wiring (Fig 3A, Table B in S1 Text). By defining the strongest axis of variation for each cell type in the wiring space, we discovered distinct spatial gradients of the cell types, which together depicted the

Cytoarchitectural similarity and externopyramidisation

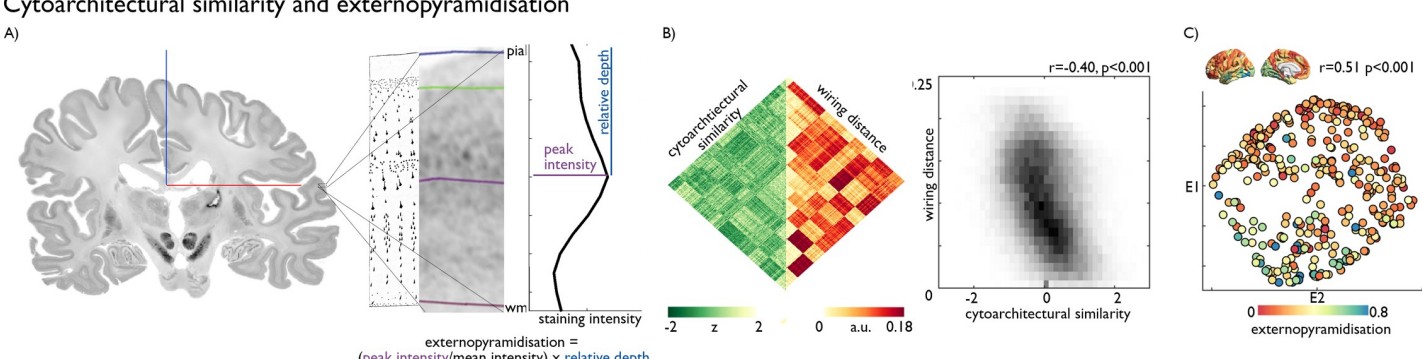

**Fig 2. Cytoarchitectural substrates of the wiring space. (A)** A 3D postmortem histological reconstruction of a human brain [35] was used to estimate cytoarchitectural similarity and externopyramidisation. Here, we present a coronal slice, a drawing of cytoarchitecture [135], magnified view of cortical layers in BigBrain and a staining intensity profile with example of calculation of externopyramidisation [44]. **(B)** Matrix and density plot depict the correlation between BigBrain-derived cytoarchitectural similarity and wiring distance between pairs of regions. **(C)** Externopyramidisation, projected onto the cortical surface and into the wiring space, is highest at the bottom of the structural manifold. Replication with the Freesurfer-style preprocessing pipeline in S5 Fig. Essential data are available on https://git.io/JTg1l.

Cell-type diversification describes variation in cortical wiring

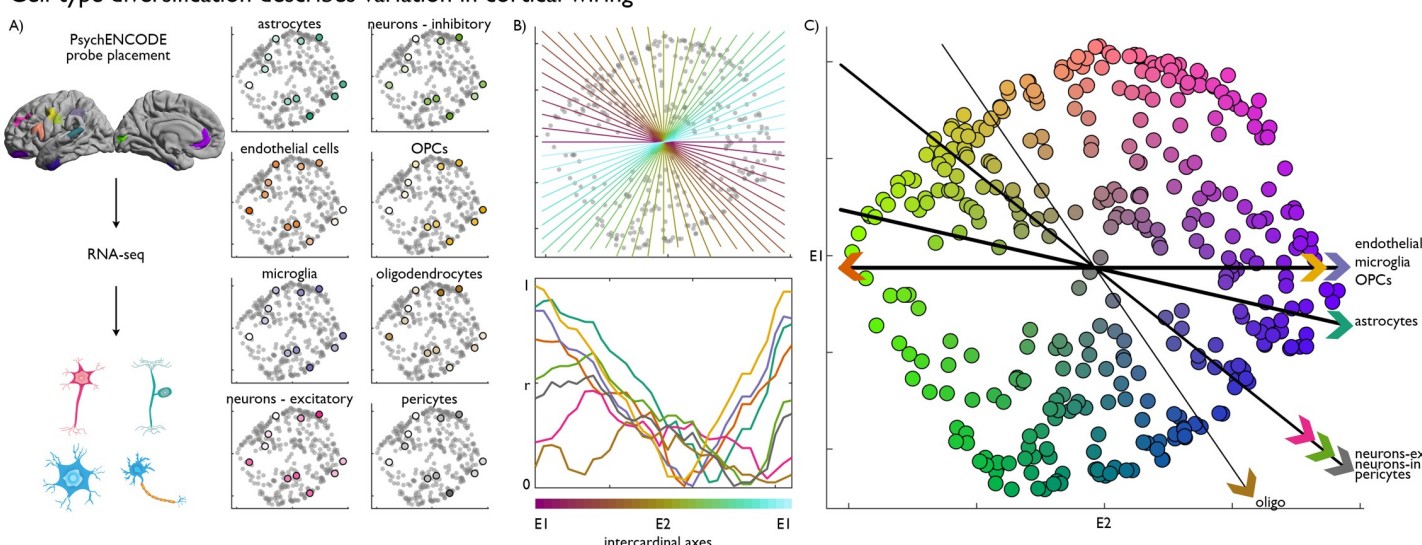

**Fig 3. Transcriptomic substrates of the wiring space. (A)** mRNA-seq probes, assigned to 11 representative nodes (coloured as in Fig 1C, i.e., their position in the wiring space), provided good coverage of the space and enabled characterisation of cell type–specific gene expression patterns. Average cell type–specific gene expression patterns projected in the wiring space, with brighter colours signifying higher expression. **(B)** Equally spaced intercardinal axes superimposed on the wiring space, and below, line plots showing correlation of gene expression patterns with each of the axes. Colours correspond to the cell types shown in part A. **(C)** Strongest axis of variation (i.e., maximum |r|) in expression of each cell type overlaid on the structural manifold. Replication with the Freesurfer-style preprocessing pipeline in S6 Fig. Essential data are available on https://git.io/JTg1l. E, eigenvector; mRNA-seq, mRNA sequencing; OPC, oligodendrocyte progenitor cell; RNA-seq, RNA sequencing.

multiform cellular differentiation of the wiring space (Fig 3B and 3C). These findings established that the wiring space captures the organisation of neuronal and nonneuronal cells (**Fig 3C**) and offers new evidence on the heightened expression of neuromodulatory glia, such as astrocytes and microglia [45, 46], towards the transmodal areas.

## Constraining role for functional architecture

Thus far, our analysis indicates that the wiring space successfully captures macroscale spatial trends in cortical organisation and that it reflects underlying cytoarchitectonic and cellular microcircuit properties. Next, we tested the hypothesis that our wiring space also underpins the functional architecture of the brain.

First, we mapped previously established intrinsic functional communities [47] into the wiring space and inspected their relative wiring distances (Fig 4A–4C, S8A–S8C Fig). Sensory networks were primarily located in the lower (visual) and upper (somatomotor) left extremities of the new coordinate system, concordant with their distinctive connectivity profiles and more specialised functional roles. In contrast, transmodal default, frontoparietal, and limbic networks were located more towards the right extremities in the wiring space. Dorsal and ventral attention networks subsumed intermediary position in the wiring space. Together, these analyses demonstrate how the relative positioning of functional communities in the structural wiring space relates to the segregation of functional networks.

Secondly, we assessed whether our wiring space can also predict macroscale functional brain connectivity. We used a supervised machine learning paradigm that applies adaptive boosting to predict functional connectivity based on relative distances of nodes in the wiring space (Fig 4D, S9 Fig; see Methods section). The new space, trained on the "Discovery" dataset, predicted resting-state functional connectivity in the independent "Hold-out" sample with

## Functional architecture reflected in the structural manifold

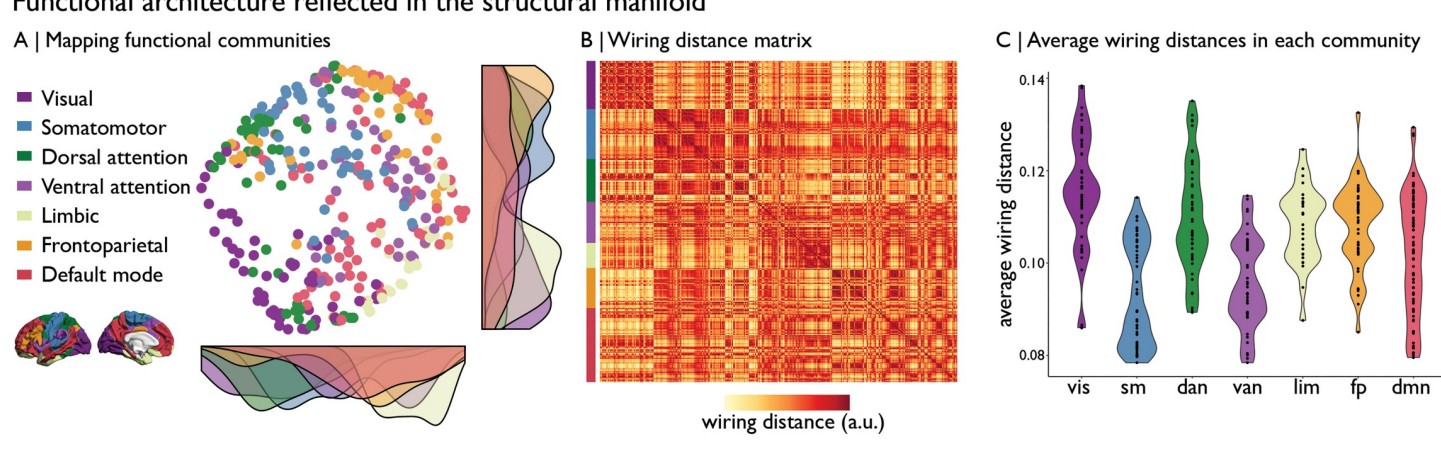

A | Mapping functional communities

- Visual
- Somatomotor
- Dorsal attention
- Ventral attention
- Limbic
- Frontoparietal
- Default mode

B | Wiring distance matrix

wiring distance (a.u.)

C | Average wiring distances in each community

average wiring distance

vis  sm  dan  van  lim  fp  dmn

## Prediction of functional connectivity in an independent sample

D | Model implementation

Discovery sample

Hyperparameter optimisation (cross-validated)

Fit model

Hold-out sample

FC  WD  ΔE1  ΔE2

predicted FC

empirical FC

E | Accuracy of predictive model - mean squared error (mse)

E1

E2

mse

F | Model comparison

| features | | | fusion | embedding | | | learner | |
|---|---|---|---|---|---|---|---|---|
| GD | MPC | TS | | E1 | E2 | E3 | linear | ML |
| X | | | | | | | X | |
| | X | | | | | | X | |
| | | X | | | | | X | |
| X | X | X | X | | | | X | |
| X | | | X | X | X | | X | |
| | X | | X | X | X | | X | |
| | | X | X | X | X | | X | |
| X | X | | X | X | X | | X | |
| X | | X | X | X | X | | X | |
| | X | X | X | X | X | | X | |
| X | X | X | X | X | X | | X | |
| X | X | X | X | X | X | X | X | |
| X | | | X | X | X | | | X |
| | X | | X | X | X | | | X |
| | | X | X | X | X | | | X |
| X | X | | X | X | X | | | X |
| X | | X | X | X | X | | | X |
| | X | X | X | X | X | | | X |
| X | X | X | X | X | X | | | X |
| X | X | X | X | X | X | X | | X |

mean squared error

G | Predictive accuracy in each community

mse

vis  sm  dan  van  lim  fp  dmn

**Fig 4. From cortical wiring to functional connectivity. (A)** Nodes in the wiring-derived coordinate system coloured by functional community [47], with the distribution of networks shown by density plots along the axes. **(B)** Wiring distance between nodes, ordered by functional community, revealed a modular architecture. **(C)** Violin plots show the average wiring distance for nodes in each functional community, with higher values being more specialised in their cortical wiring. **(D)** Using the boosting regression models from the "Discovery" dataset, we used features of the wiring space to predict z-standardised functional connectivity in a "Hold-out" sample. The model was enacted for each node separately. **(E)** MSE across nodes are shown in the wiring space and on the cortical surface (Table C in S1 Text). **(F)** Predictive accuracy of various cortical wiring models, involving the use of different features, multifeature fusion, eigenvectors from diffusion map embedding, and a linear or ML learner. **(G)**

MSE of the wiring space model stratified by functional community. Replication with the Freesurfer-style preprocessing pipeline in S8 Fig. Essential data are available on https://git.io/JTg1l. ΔE1, difference on eigenvector 1; ΔE2, difference on eigenvector 2; a.u., arbitrary units; FC, functional connectivity; ML, machine learning; MSE, mean squared error; WD, wiring distance.

high accuracy (mean squared error = 0.49 ± 0.16; $R^2$ = 0.50 ± 0.16; Fig 4E) and outperformed learners trained on data from fewer cortical wiring features or learners trained on all modalities but without using manifold embedding (Fig 4F, Table C in S1 Text). Inspecting regional variations in predictive accuracy indicated that cortical wiring topography was more tightly linked to functional connectivity in sensory areas, systems upon which classical examples of the cortical hierarchy were developed [48], while it tapered off towards transmodal cortex. Further expanding the wiring space to 3 eigenvectors/dimensions did not substantially improve model performance (mean squared error = 0.51 ± 0.15; $R^2$ = 0.56 ± 0.22, Table C in S1 Text). While accuracy was reduced, the 2D model also provided state-of-the-art predictions of resting-state functional connectivity in individual participants of the "Hold-out" dataset (mean squared error = 1.32 ± 0.35, $R^2$ = 0.16 ± 0.12; S3B Fig).

High predictive performance at the group level could be replicated in the independent MICs dataset, despite the smaller sample size (mean squared error = 0.40 ± 0.11, $R^2$ = 0.59 ± 0.12; S2B Fig).

## Large-scale organisation of directed coherence

The above analyses showed that the wiring space robustly explains aspects of macroscale functional organisation and connectivity. We next examined whether it can also account for a more direct measure of neural functional connectivity, by examining stereo-electroencephalographic recordings during resting wakeful rest in ten epileptic patients (who underwent multimodal imaging before the implantations, with imaging identical to the MICs dataset; Fig 5A). Patients presented with a similar wiring space solution as controls from the same sample (S10 Fig; correlations between eigenvectors 1/2: r = 0.92/0.89). In line with the above functional connectivity analysis, the wiring model explained substantial within sample variance in undirected coherence ($R^2$ = 0.66 ± 0.23; Fig 5B, S11 Fig), especially in frequencies >18 Hz (0.61 < $\bar{R}^2$ < 0.76). We calculated the phase slope index within frequency bands as an estimate of unidirectional flow [49]. To highlight large-scale organisation and account for the incomplete coverage of electrodes in each participant, we clustered the wiring space into 12 macroscopic compartments (Fig 5C; see Methods section for determination of k = 12). We estimated the phase slope index between each pair of clusters and performed significance testing using a linear mixed-effect model that included participant as a random effect. Decomposition of intercluster similarities in the phase slope index using a principle component analysis revealed a gradient running across the wiring space. The first principle component, accounting for 39% of variance, illustrated a transition in the patterns of directed coherence from the upper left to lower right of the wiring space, which is running from central to temporal and limbic areas (Fig 5D). The component loading was underpinned by varied expression of cell types (Fig 5D). Lasso regularisation showed that inhibitory neuron expression was the most important cellular feature in supporting this coherence-derived topography, followed by microglia in highly regularised models. Together, inhibitory neuron and microglia cell expression accounted for 41% of variance in component loading ($p$ = 0.05). The cell types did not reach significance in explaining the component loading independently, emphasising the multivariate contribution of cell types to spatial variations in electrophysiological oscillations. Robustness of the component loading and edge-wise phase slope index estimates were supported by a leave-one-subject-out procedure (S12 Fig).

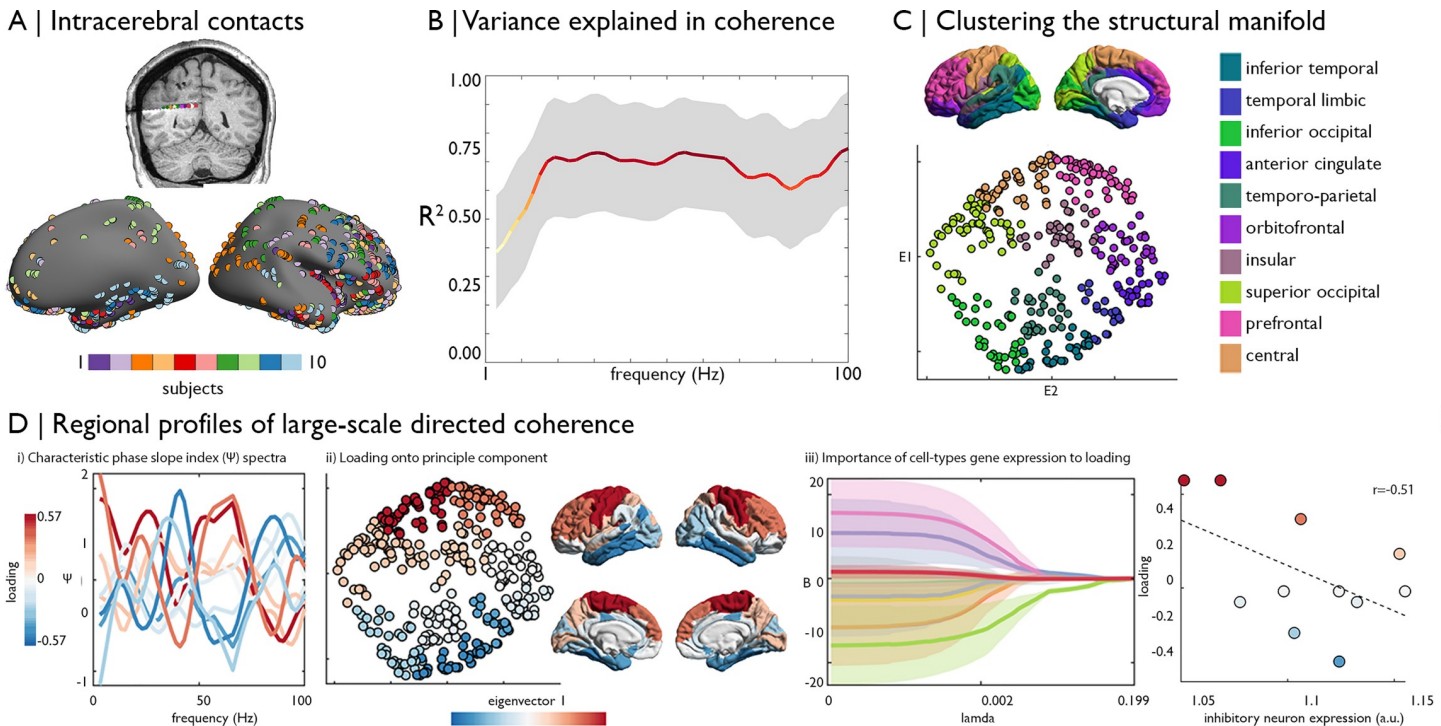

**Fig 5. Large-scale organisation of coherence in the structural manifold. (A)** Intracerebral implantations of 10 epileptic patients were mapped to the cortical surface and intracortical EEG contacts selected. We studied 5 minutes of wakeful rest. **(B)** Mean and standard deviation in the variance explained in undirected coherence by wiring space features using adaboost machine learning across all nodes. **(C)** Clusters of the wiring space. **(D)** Phase slope index ($\Psi$) was calculated for each pair of intra-subject electrodes, then cluster-to-cluster estimates were derived from a linear mixed-effect model. Pearson correlation across $\Psi$ estimates was used to measure the similarity of clusters, and the major axis of regional variation was identified via principle component analysis. (i) Average $\Psi$ spectra for each region coloured by loading on the first principle component (accounting for 39% of variance). (ii) Component loadings presented in the wiring space and on the cortical surface illustrate a gradient from upper left regions, corresponding to central areas, towards lower right areas, corresponding to temporal and limbic areas. (iii) Lasso regularisation demonstrates the contribution of cell type–specific gene expression (colours matching Fig 3) and externopyramidisation (red) to explain variance in the component loadings. Shaded areas show the standard deviation in fitted least-squares regression coefficients across leave-one-observation-out iterations. For example, inhibitory neurons expression levels (green) are closely related to the component loading, as shown in the scatterplot. Replication with the Freesurfer-style preprocessing pipeline in S11 Fig. Essential data are available on https://git.io/JTg1l. a.u., arbitrary units; EEG, electroencephalography.

Finally, we examined the topology of large-scale networks of directed coherence in frequencies influencing the component loading and specifically tested whether they met the criteria for hierarchical organisation [48] (Fig 6, S13 Fig). Both the 22 to 26 Hz (beta) and 84 to 88 Hz (high gamma) bands met criteria for hierarchy, insomuch that clusters could be placed in levels that depict unidirectional flow of oscillations from the top to bottom of the graph. The beta band was associated with 2 anterior–posterior streams, whereas high gamma band was related to a looping wave from inferior temporal cortex through occipital and parietal regions to the prefrontal cortex. These results support the hierarchical organisation of large-scale directed coherence, which propagate as waves of oscillations moving through the cortical wiring scheme, and also demonstrate the co-occurrence of hierarchies and that these are operationalised in different frequencies.

## Discussion

Based on advanced machine learning of multiple features sensitive to cortico–cortical wiring, our work identified a novel and compact coordinate system of human cortex. Our analysis established that cortical wiring is dominated by 2 principal axes, 1 running from sensory

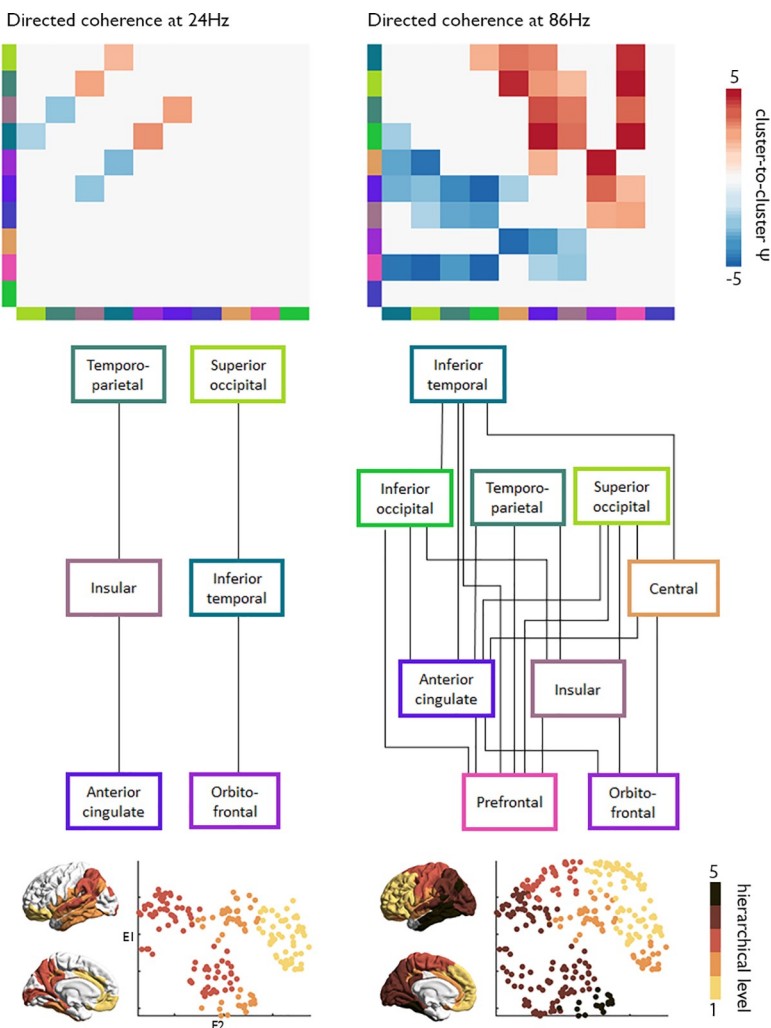

**Fig 6. Hierarchical information processing is organised within the structural manifold.** "Top": The most influential frequencies on the component loading were identified through a correlation of average Ψ spectra (Fig 5Di) with the component loading. For the global maxima (24 Hz, left) and minima (86 Hz, right), we present cluster-to-cluster Ψ estimates, thresholded at $p < 0.05$. "Bottom": Suprathreshold edges are plotted as a hierarchical schema, where the directed coherence estimates indicate flow of oscillations from the top to the bottom of the hierarchy. The hierarchical level of each cluster is also presented on the cortical surface and in the wiring space, illustrating unique spatial patterns of the frequency-specific hierarchies. Replication with the Freesurfer-style preprocessing pipeline in S13 Fig. Essential data are available on https://git.io/JTg1l.

towards transmodal systems and 1 running from anterior to posterior. Critically, this novel space successfully accounted for both local descriptions of cortical microcircuitry as well as macroscale cortical functional dynamics measured by functional MRI and intracranial electrical recordings. By projecting postmortem histological and transcriptomic profiles into this newly discovered space, we could demonstrate how these axes are determined by intersecting cell type–specific and cytoarchitectural gradients. In addition to establishing these local neurobiological features, our findings support that the wiring space serves as a powerful scaffold within which macroscale intrinsic human brain function can be understood. Using both non-invasive imaging in healthy individuals and direct neuronal measurements in a clinical population, we demonstrated that the wiring-derived manifold describes how neural function is hierarchically organised in both space and time. Our findings were replicable in different

datasets and at the single-subject level; moreover, a series of additional experiments showed that this novel representation outperformed conventional approximations of structural connectivity in their ability to predict function. Together, we have successfully identified a compact description of the wiring of the cortex that can help to ultimately understand how neural function is simultaneously constrained by both local and global features of cortical organisation.

Our multivariate model of cortical wiring reflects a useful extension on diffusion MRI tractography because it additionally incorporates spatial proximity and microstructural similarity––2 features tapping into generative principles of cortico–cortical connectivity as demonstrated by prior human and nonhuman animal studies [30–33]. The application of manifold learning to this enriched representation of cortical wiring helped to determine a low dimensional, yet highly expressive, depiction of cortical wiring. In other fields, notably genomics and data science, more generally, embedding techniques have become widely adopted to identify and represent the structure in complex, high-dimensional datasets [36, 38, 50, 51]. In recent neuroimaging studies, several approaches have harnessed nonlinear dimensionality reduction techniques to identify manifolds from single modalities, highlighting changes in microstructure and function at the neural system level [7, 34, 52, 53]. The wiring space identified here captured both sensory-fugal and anterior–posterior processing streams, 2 core modes of cortical organisation and hierarchies established by seminal tract-tracing work in nonhuman primates [54, 55]. The anterior–posterior axis combines multiple local gradients and functional topographies, such as the ventral visual stream running from the occipital pole to the anterior temporal pole that implements a sensory–semantic dimension of perceptual processing [56, 57] and a rostro-caudal gradient in the prefrontal cortex that describes a transition from high-level cognitive processes supporting action preparation to those tightly coupled with motor execution [55, 58–60]. The sensory-fugal axis represents an overarching organisational principle that unites these local processing streams. While consistent across species, the number of synaptic steps from sensory to higher-order systems has increased throughout evolution, supporting greater behavioural flexibility [54] and decoupling of cognitive functioning from the here and now [57]. By systematically studying internodal distances within the wiring-derived space, it may be possible to gain a more complete understanding of the difference between specialised systems at the periphery and more centrally localised zones of multimodal integration, such as the temporo–parietal junction and cingulate cortex. Many of these regions have undergone recent evolutionary expansion [61], are sites of increased macaque–human genetic mutation rates [62], and exhibit the lowest macaque–human functional homology [63]. In this context, the more complete model of cortical wiring provided here may play a critical role in advancing our understanding of how changes in cortical organisation have given rise to some of the most sophisticated features of human cognition.

Our new coordinate system reflects an intermediate description of cortical organisation that simultaneously tracks its microstructural underpinnings and also addresses the emergence of functional dynamics and hierarchies at the system level. Cross-referencing the new space with a 3D histological reconstruction of a human brain [35] established close correspondence between our in vivo model and histological measurements. This work adds to the notion that cytoarchitecture and cortical wiring are inherently linked [64, 65], as variations in projections across cortical layers determine the layout of the cortical microcircuitry [66]. Feedforward connections often originate in supragranular layers (and terminate in lower layers in the target regions), while infragranular layers give rise to feedback projections flowing down the cortical hierarchy [67–70]. In this work, we observed an alignment between the cortical wiring scheme and a proxy for externopyramidisation [44], which was sensitive to inter-areal differences in the depth of peak pyramidal neuron density that co-occurs with shifts from feedback to

feedforward dominated connectivity profiles [5]. In addition to showing cytoarchitectonic underpinnings of the cortical microcircuitry, we capitalised on postmortem gene expression datasets derived from mRNA sequencing [71], a technique thought to be more sensitive and specific than microarray transcriptomic analysis. This approach revealed that divergent gradients of cell type–specific gene expression underpin intercardinal axes of the new coordinate system, particularly of nonneuronal cell types. Increased glia-to-neuron ratios in transmodal compartments of the new space may support higher-order cognitive functions, given comparative evidence showing steep increases in this ratio from worms to rodents to humans [72–75]. Astrocytes, in particular, exhibit morphological variability that may lend a cellular scaffold to functional complexity and transmodal processing [72]. For example, a uniquely human interlaminar astrocyte was recently discovered with long fibre extensions, likely supporting long-range communication between distributed areas that may contribute to flexible, higher-order cognitive processing [76].

Our coordinate system also established how structural constraints relate to cortical dynamics and information flow throughout hierarchical and modular systems. We showed that functional communities are circumscribed within the wiring-derived space, supporting dense within-network connectivity, and that their relative positions describe a progressive transition from specialised sensory wiring to an integrative attentional core and distributed transmodal networks. Region-to-region distances in the wiring space provided competitive predictions of resting-state functional connectivity data, both at the level of the group and of a single participant. Spatial proximity and microstructural similarity are critical elements of the predictive value of our models, highlighting intracortical and cytoarchitecturally matched projections in shaping intrinsic functional organisation. However, these dominant aspects of cortical wiring are often not considered by computational models that simulate functional connectivity from measures of diffusion-based tractography [17, 18, 77]. In addition to feature enrichment, the use of nonlinear dimensionality reduction enhanced the predictive performance by minimising the influence of noisy edges and magnifying effects from the most dominant axes of cortical wiring. Such a model led to maximal gains in predictive power in unimodal areas, likely owing to their more locally clustered and hierarchically governed connectivity profiles [78–80]. Predictive performance decreased towards transmodal networks, a finding indicating that more higher-level systems may escape (currently measurable) structural constraints and convergence of multisynaptic pathways [81]. Such conclusions are in line with recent work showing that transmodal areas exhibit lower microstructure-function correspondence [7] and reduced correlations between diffusion tractography and resting-state connectivity [82, 83], potentially contributing to greater behavioural flexibility [7, 84]. The hierarchical nature of the wiring space was further supported by analysing its correspondence to direct measurements of neural dynamics and information flow via intracerebral stereo-electroencephalography. Using a strict definition of hierarchy as the topological sequence of projections [48, 85], we demonstrated that the wiring space underpins large-scale, frequency-specific waves of oscillations propagating from anterior to posterior and from limbic to prefrontal. Furthermore, bridging across scales, we provide an additional line of evidence for the importance of inhibitory neurons for supporting specific frequencies of oscillations, such as the role of somatostatin for beta oscillations [86]. One key feature of the present framework, therefore, is that it provides a basis to quantitatively assess how the interplay of neuronal oscillations underpin complex cortical organisation. Even with limited spatial resolution, data-driven decomposition of the wiring space proved an effective model of electrophysiological organisation, in line with recent work showing the modular architecture of electrocorticography [87].

Future studies should increase the resolution of the wiring space, which may be possible with ongoing efforts to generate robust estimates of white matter tracts from single voxels [88].

Efforts will also benefit from the incorporation of subcortical nodes, such as the thalamus, which are increasingly recognised to play major roles in cortico–cortical dynamics and have distinct cell populations projecting to specific cortical laminae [89, 90]. In lieu of a gold standard for cortical wiring in humans, the present work focused on equally balanced cortical wiring features; however, supervised learning techniques could reveal their relative importance for specific tasks or scales. As our results have shown, this novel representation of cortical wiring provides a practical workspace to interrogate the coupling of brain structure and function and to study links between microcircuit properties and macroscale hierarchies. As such, the wiring space can be a powerful tool to study the multi-scale complexities of brain development, ageing, and disease.

## Methods

### Human connectome project dataset

**Data acquisition.** We studied 197 unrelated healthy adults from the S900 release of the HCP [93]. The "Discovery" dataset included 100 individuals (66 females, mean ± SD age = 28.8 ± 3.8 years) and the "Hold-out" dataset included 97 individuals (62 females, mean ± SD age = 28.5 ± 3.7 years). MRI data were acquired on the HCP's custom 3T Siemens Skyra (Siemens AG, Erlanger, Germany) equipped with a 32-channel head coil. Two T1w images with identical parameters were acquired using a 3D-MPRAGE sequence (0.7-mm isotropic voxels, matrix = 320 × 320, 256 sagittal slices; TR = 2,400 ms, TE = 2.14 ms, TI = 1,000 ms, flip angle = 8˚; iPAT = 2). Two T2w images were acquired using a 3D T2-SPACE sequence with identical geometry (TR = 3,200ms, TE = 565 ms, variable flip angle, iPAT = 2). A spin-echo echo-planar imaging (EPI) sequence was used to obtain diffusion-weighted images, consisting of 3 shells with $b$-values 1,000, 2,000, and 3,000 s/mm$^2$ and up to 90 diffusion weighting directions per shell (TR = 5,520 ms, TE = 89.5 ms, flip angle = 78˚, refocusing flip angle = 160˚, FOV = 210 × 180, matrix = 178 × 144, slice thickness = 1.25 mm, mb factor = 3, echo spacing = 0.78ms). Four resting-state functional magnetic resonance imaging (rs-fMRI) scans were acquired using multiband accelerated 2D-BOLD EPI (2-mm isotropic voxels, matrix = 104 × 90, 72 sagittal slices, TR = 720 ms, TE = 33 ms, flip angle = 52˚, mb factor = 8, 1,200 volumes/scan, 3,456 seconds). Participants were instructed to keep their eyes open, look at fixation cross, and not fall asleep. Nevertheless, some participants were drowsy and may have fallen asleep [91], and the group averages investigated in the present study do not address these interindividual differences. While T1w, T2w, and diffusion scans were acquired on the same day, rs-fMRI scans were split over 2 days (2 scans/day).

**Data preprocessing.** MRI data underwent HCP's minimal preprocessing [93]. Cortical surface models were constructed using Freesurfer 5.3-HCP [94–96], with minor modifications to incorporate both T1w and T2w [97]. Following intensity nonuniformity correction, T1w images were divided by aligned T2w images to produce a single volumetric T1w/T2w image per participant, a contrast ratio sensitive to cortical microstructure [97]. Diffusion MRI data underwent correction for geometric distortions and head motion [93]. BOLD time series were corrected for gradient nonlinearity, head motion, bias field, and scanner drifts, then structured "noise" components were removed using ICA-FIX, further reduce the influence of motion, nonneuronal physiology, scanner artefacts, and other nuisance sources [98]. The rs-fMRI data were resampled from volume to MSMAll functionally aligned surface space [99, 100].

### Generation of the wiring features

Cortical wiring features were mapped between spatially contiguous cortical "nodes." We advanced 2 approaches throughout all analyses, 1 based on multimodal parcellations

recommended by the HCP ("HCP-style") and 1 based on sulco-gyral parcellations implemented in FreeSurfer ("FreeSurfer-style"). For HCP-style, a 360 node parcellation scheme was used, in which areal boundaries were defined by gradients of change in mean T1w/T2w intensity, cortical thickness, task-related BOLD activation, and/or resting-state functional connectivity [101]. For Freesurfer-style, a 200 node, semi-random division of the Desikan Killany atlas [102] was used, with approximately equal sized nodes. By providing both approaches, we aim to demonstrate that our hypotheses may be supported by either style, although we did not set out to compare their performance.

**Geodesic distance.** GD was calculated across subject-specific mid-cortical surface maps in native space. Exemplar vertices of each node were defined for each participant as the vertex with minimum Euclidean distance to the subject-specific node centroid. For HCP-style, we used workbench commands to calculate the GD between exemplar vertices by travelling along the cortical midsurface [26, 27, 34]. To compensate for the inability of such an approach to cross hemispheres, we mirrored the GD. For FreeSurfer-style, we adopted a recently published GD calculation approach that combines surface and volume-based GD calculations [103]. For each node, we matched the exemplar vertex to the nearest voxel in volumetric space, and then used a Chamfer propagation (imGeodesics Toolbox; https://github.com/mattools/matImage/wiki/imGeodesics) to calculate the distance to all other voxels travelling through a grey/white matter mask. This approach differs from cortex-constrained GD implemented in workbench [26, 27, 34] by involving paths through the grey and white matter, allowing for jumps within gyri and interhemispheric projections [103] and thereby potentially tracking intracortical, as well as short-range association fibres [104]. We projected GD estimations back from volumetric to surface space, averaged within node, and produced a symmetric GD matrix.

**Microstructure profile covariance.** The full procedure of the MPC approach may be found elsewhere [7]. In brief, we generated 12 equivolumetric surfaces between the outer and inner cortical surfaces [105, 106] and systematically sampled T1w/T2w values along linked vertices across the whole cortex. Even though the number of surfaces exceeds the maximum number of voxels within the cortical ribbon, this slight oversampling confers high stability of the resultant MPC matrix [7]. T1w/T2w intensity profiles were averaged within nodes, excluding outlier vertices with median intensities more than 3 scaled median absolute deviations away from the node median intensity. Nodal intensity profiles underwent pairwise Pearson product–moment correlations, controlling for the average whole-cortex intensity profile. The MPC matrix was absolutely thresholded at 0; remaining MPC values were then log-transformed to produce a symmetric MPC matrix.

**Tractography strength.** Tractographic analysis was based on MRtrix3 (https://www.mrtrix.org). Response functions for each tissue type were estimated using the dhollander algorithm [107]. Fibre orientation distributions (i.e., the apparent density of fibres as a function of orientation) were modelled from the diffusion-weighted MRI with multi-shell multi-tissue spherical deconvolution [108], then values were normalised in the log domain to optimise the sum of all tissue compartments towards 1, under constraints of spatial smoothness. Anatomically constrained tractography was performed systematically by generating streamlines using second order integration over fibre orientation distributions with dynamic seeding [109, 110]. Streamline generation was aborted when 40 million streamlines had been accepted. Using a spherical-deconvolution informed filtering of tractograms (SIFT2) approach, interregional TS was taken as the streamline count weighted by the estimated cross section [110]. A group-representative TS matrix was generated using distance dependent consensus thresholding [111]. The approach involves varying the consensus threshold as a function of distance. The resulting connectivity matrix preserves the pooled edge length distribution of subject-level data as well

as integral organisational features, such as long-range connections, while reducing false-positive edges. The group-representative matrix contained 13.7% of possible edges.

## Correspondence of cortical wiring features

To assess the complementarity of these features in characterising cortical wiring, we computed matrix-wide Spearman correlations between all feature pairs (MPC-GD, MPC-TS, and GD-TS). We also assessed regional variations in feature correspondence at each node using Spearman correlations and the standard deviation in a multifeature fingerprint (S2 Fig).

## Building the wiring space

**Overview of approach.** The wiring space was built through the integration of MPC, GD, and TS. In an effort to provide our community access to the methods we used here, we have made normative manifold maps openly available (https://github.com/MICA-MNI/micaopen/tree/master/structural_manifold) and incorporated all relevant functions and workflow into the BrainSpace toolbox (http://brainspace.readthedocs.io; [112]). The procedure is as follows:

i. **Normalisation:** Nonzero entries of the input matrices were rank normalised. Notably, rank normalisation was performed on the inverted form of the GD matrix, i.e., larger values between closer regions. The less sparse matrices (GD and MPC) were rescaled to the same numerical range as the sparsest matrix (TS) to balance the contribution of each input measure.

ii. **Fusion:** Horizontal concatenation of matrices and production of a node-to-node affinity matrix using row-wise normalised angle similarity. The affinity matrix thus quantifies the strength of cortical wiring between 2 regions. Alternative data fusion techniques, such as similarity network fusion [113] and joint embedding [63], aim to identify similar motifs across modalities. A key outcome of those approaches is higher signal-to-noise ratio; however, unique network information provided by each modality would be minimised. Given that our cross-modal structural analyses highlighted modality-specific principles of cortical organisation, we sought to use the concatenation approach that preserves distinct information in each modality.

iii. **Manifold learning:** Diffusion map embedding was employed to gain a low-dimensional representation of cortical wiring. Diffusion map embedding belongs to the family of graph Laplacians, which involve constructing a reversible Markov chain on an affinity matrix. Compared to other nonlinear manifold learning techniques, the algorithm is robust to noise and computationally inexpensive [114, 115]. A single parameter α controls the influence of the sampling density on the manifold (α = 0, maximal influence; α = 1, no influence). As in previous studies [34, 116], we set α = 0.5, a choice retaining the global relations between data points in the embedded space. Notably, different alpha parameters had little to no impact on the first 2 eigenvectors (spatial correlation of eigenvectors, r > 0.99). We operationalised a random walker to approximate the likelihood of transitions between nodes, illuminating the local geometry in the matrix. Preservation of local geometry using the kernel critically differentiates diffusion maps from global methods, such as principle component analysis and multidimensional scaling. Local geometries are integrated into a set of global eigenvectors by running the Markov chain forward in time. The decay of an eigenvector provides an integrative measure of the connectivity between nodes along a certain axis. This lower-dimensional representation of cortical wiring is especially interesting for interrogating the cortical hierarchy, which previous research suggests extends upon sensory-fugal and anterior–posterior axes. In the present study, the

number of dimensions selected for further analysis was determined based on the variance explained by each eigenvector, where the cutoff point determined using the Cattell scree test. This resulted in 2 dimensions, which aligns with the hypothesised number of axes and, fortunately, can be readily visualised. Furthermore, the first 2 eigenvectors were robust across alternative dimensionality reduction approaches, such as linear principle component analysis (r > 0.99) and nonlinear Laplacian eigenmaps (r > 0.99).

**Key outcome metrics.** The wiring space represents the principle axes of variation in cortical wiring, as well as their interaction. We displayed the conversion from anatomical to wiring space using a 3-part colourmap. The colour of each node was ascribed based on proximity to the limits of the wiring space: blue for closeness to the maximum of the second eigenvector, green for closeness to minimum of the second eigenvector, and redness represents closeness to maximum of the first eigenvector.

The relative positioning of nodes in the wiring space informs on the strength of cortical wiring along the principle axes. We characterised the relative positioning of each pair of nodes with wiring distance and difference along each primary axis, which pertain to node-to-node proximity and axis-specific shifts, respectively. To calculate wiring distances, we triangulated the wiring space used a Delaunay approach and calculated GD between each node used the Fast Marching Toolbox (https://github.com/gpeyre/matlab-toolboxes/tree/master/). The average wiring distance of each node informs upon centrality within the space and reflects a region's propensity to have many cortical connections.

## Neurobiological substrates of the wiring space

**Association to cytoarchitectural features.** For the cytoarchitectonic maps, a 100-μm resolution volumetric histological reconstruction of a postmortem human brain from a 65-year-old male was obtained from the open-access BigBrain repository [35] on February 2, 2018 (https://bigbrain.loris.ca/main.php). Using previously defined surfaces of the layer 1/11 boundary, layer 4, and white matter [117], we divided the cortical mantle into supragranular (layer 1/11 to layer 4) and infragranular bands (layer 4 to white matter). Staining intensity was sampled along 5 equivolumetric surfaces within the predefined supra- and infragranular bands at 163,842 matched vertices per hemisphere, then averaged for each parcel. We estimated cytoarchitectural similarity of regions by performing the above MPC procedure on BigBrain-derived intracortical profiles, as in previous work [7]. Externopyramidisation [44], described as the "gradual shift of the weight of the pyramidal layers from the V to the IIIc," was approximated as the product of the normalised peak intensity and the relative thickness of the supragranular layers, i.e.,

$$\text{Externopyramidisation} = \frac{\max(intensity)}{\text{mean}(intensity)} \times \frac{1 - thickness_{supra}}{thickness_{total}}$$

Intensity and thickness quotients were independently rescaled between 0 and 1 across all regions to balance their contribution to the externopyramidisation metric. Higher values reflect higher intensity values and shallower depth of the peak layer.

**Cell type–specific gene expression.** Cell type–specific gene lists were derived from an analysis of >60,000 single cells extracted from human adult visual cortex, frontal cortex, and cerebellum with single-nucleus Droplet-based sequencing (snDrop-seq) or single-cell transposome hypersensitive-site sequencing (scTHS-seq) [118]. We focused on 8 canonical cell classes: astrocytes, endothelial cells, microglia, inhibitory neurons, excitatory neurons, oligodendrocytes, oligodendrocyte progenitor cells (OPCs), and pericytes. Cell type–specific expression

maps were calculated as the average of log2 normalised gene expression across 11 neocortical areas in 12 human adult brains [36, 43]. Areas were visually matched to the nearest parcel. Interregional co-expression was calculated as the inverse of Euclidean distance between cell type–specific gene expression.

The influence of neurobiological similarities on relative positioning of nodes in the wiring space was tested by performing Spearman correlations of wiring distance with cytoarchitectural similarity and cell type–specific gene co-expression patterns, with and without controlling for GD. We used multiple linear regressions to evaluate the variance explained in externopyramidisation and cell type–specific expression by the 2 wiring space eigenvectors. Significance values were corrected to account for spatial autocorrelation in the eigenvectors using spin testing and Moran spectral randomisation, respectively [119, 120], and was operationalised using BrainSpace [112]. Spectral randomisation was initialised using the GD matrix. As an additional control analysis, we tested the correspondence of the first 2 eigenvectors with features of cortical morphometry using the same procedure.

The wiring space offers a dimensional approach to evaluate the concordance of gradients at multiple biological scales. To facilitate multi-scale comparisons, we generated 32 axes within the wiring space by creating inter-cardinal lines in 5.625˚ steps. Linear polynomial equations, corresponding to each inter-cardinal line, were evaluated for 100 equally spaced x and y-values between the minimum and maximum range of the first and second eigenvectors, respectively. Nodes were assigned the values of the nearest point along each inter-cardinal line, based on Euclidean distance, thus the position of a node on each axis could be represented by an integer [1–100]. The dominant axis of variation of any feature in wiring space can be classified as the axis of maximum correlation. *p*-values from the Spearman correlations were subjected to false discovery rate (FDR) correction to assess whether the dominant axis of variation was significant [121].

## Association with functional MRI-based connectivity

**Functional architecture.**  The wiring space was reimagined as a completely dense network with edges weighted by wiring distance (Fig 3B). We mapped 7 established functional communities [47] into the group-level wiring space by assigning each node to the functional community that was most often represented by the underlying vertices.

**Predicting functional connectivity.**  Individual functional connectomes were generated by averaging preprocessed time series within nodes, correlating nodal time series, and converting them to z scores. For each individual, the 4 available rs-fMRI scans were averaged at the matrix level, then the connectomes were averaged within the "Discovery" and "Hold-out" samples separately. We estimated the variance explained in functional connectivity by the cortical wiring scheme using boosted regression trees [122]. Boosted regression trees produce a predictive model from the linear weighted combination of weaker base learners that each fit the mean response of a subsection of the predictor space. Weak estimators are built in a step-wise manner, with increasing focus on poorly explained sections of the predictor space. Optimisation of the learning rate and number of estimators is critical to model complex nonlinear relationships, implicitly model interactions between predictors, and reduce overfitting. Overfitting was further reduced and predictive performance enhanced by using a random subset of data to fit each new tree. The present study specifically used the AdaBoost module of scikit-learn v 0.21.3 in Python 3.5 and established the optimal number of estimators and the learning rate using internal 5-fold cross-validation [maximum tree depth = 4, number of estimators = 6:2:20, learning rate = (0.01, 0.05, 0.1, 0.3, 1); see S9B Fig for node-wise hyperparameters]. We aimed to predict functional connectivity independently for each node based on wiring distance,

difference along the first eigenvector, and difference along the second eigenvector to all other nodes. Each feature was z-standardised before being entered in the model. Predictive accuracy was assessed as the mean squared error and $R^2$ coefficient of determination of empirical and predicted functional connectivity. Given prior z-standardisation of features, a mean squared error of 1 would represent an error of 1 standard deviation from the true value. An $R^2$ above 0 indicates predictive value of the model, where 1.0 is the maximum possible score.

## Replication of the wiring space in an independent dataset

Independent replication was performed using locally acquired data from 40 healthy adults (MICs cohort; 14 females, mean ± SD age = 30.4 ± 6.7, 2 left-handed) for whom quantitative T1 (qT1) relaxation time mapping images were available. All participants gave informed consent, and the study was approved by the local research ethics board of the Montreal Neurological Institute and Hospital. MRI data were acquired on a 3T Siemens Magnetom Prisma-Fit (Siemens AG) with a 64-channel head coil. A submillimetric T1-weighted image was acquired using a 3D-MPRAGE sequence (0.8-mm isotropic voxels, 320 × 320 matrix, 24 sagittal slices, TR = 2,300 ms, TE = 3.14 ms, TI = 900 ms, flip angle = 9˚, iPAT = 2) and qT1 data was acquired using a 3D-MP2RAGE sequence (0.8-mm isotropic voxels, 240 sagittal slices, TR = 5,000ms, TE = 2.9 ms, TI 1 = 940 ms, T1 2 = 2,830 ms, flip angle 1 = 4˚, flip angle 2 = 5˚, iPAT = 3, bandwidth = 270 Hz/px, echo spacing = 7.2ms, partial Fourier = 6/8). The combination of 2 inversion images in qT1 mapping minimises sensitivity to B1 inhomogeneities [41] and provides high intra-subject and inter-subject reliability [123]. A spin-echo EPI sequence was used to obtain diffusion-weighted images, consisting of 3 shells with b-values 300, 700, and 2,000 s/mm$^2$ and 10, 40, and 90 diffusion weighting directions per shell, respectively (TR = 3,500 ms, TE = 64.40 ms, 1.6-mm isotropic voxels, flip angle = 90˚, refocusing flip angle = 180˚, FOV = 224 × 224 mm$^2$, slice thickness = 1.6 mm, mb factor = 3, echo spacing = 0.76ms). One 7-minute rs-fMRI scan was acquired using multiband accelerated 2D-BOLD EPI imaging (TR = 600 ms, TE = 30 ms, 3-mm isotropic voxels, flip angle = 52˚, FOV = 240 × 240 mm$^2$, slice thickness = 3 mm, mb factor = 6, echo spacing = 0.54 mms). Participants were instructed to keep their eyes open, look at fixation cross, and not fall asleep.

The data preprocessing and construction of the wiring space were otherwise virtually identical to the original HCP dataset, with a few exceptions. Microstructure profiles were sampled from qT1 images. Cortical surface estimation via FreeSurfer utilised 2 T1-weighted scans, and surface models were manually edited for accuracy. All fMRI data underwent gradient unwarping, motion correction, fieldmap-based EPI distortion correction, brain boundary-based registration of EPI to structural T1-weighted scan, nonlinear registration into MNI152 space, and grand mean intensity normalisation. The rs-fMRI data were additionally denoised using an in-house trained ICA-FIX classifier [98, 124] as well as spike regression [125, 126]. Time series were sampled on native cortical surfaces and resampled to fsaverage via folding-based FreeSurfer surface registration [96].

## Intracranial EEG analyses in epileptic patients

A group of 10 patients with drug-resistant focal epilepsy (1 male, mean ± SD age = 28.9 ± 7.9, all right-handed) were scanned using the same imaging protocol as the heathy controls from the "Replication" dataset. Patients furthermore underwent intracerebral stereo-electroencephalographic investigation as part of their presurgical evaluation, after the imaging. The protocol received prior approval from the Montreal Neurological Institute (MNI) Institutional Review Board. The recordings were acquired with Nihon Khoden EEG amplifiers (Nihon Kohden Corporation, Tokyo, Japan) at a sampling rate of 2,000 Hz, using 1 single type of depth

electrodes (DIXI electrodes with either 10 or 15 electrode). A board certified neurophysiologist (BF) selected epochs without ictal events, absent of artefacts, from periods of resting wakefulness with eyes closed during standardised conditions, resulting in 1 to 2 minutes of recording for each patient.

Each depth electrode was mapped to a cortical parcel using with the following procedure. For each participant, cortical surfaces were extracted from the high-resolution preimplantation T1-weighted using FreeSurfer6.0. Next, a clinical structural image, acquired during the implantation period on a Philips Medical Systems 1.5T MRI scanner, was transformed to the T1-weighted space using volume-based affine transformation with nearest neighbour interpolation. For 7 patients, the clinical scan was a T1-weighted image (3D SENSE, slice thickness = 0.78 mm, number of slices = 280, single-echo, phase-encoding steps = 320, echo train length = 320, TR = 0.0079 s, flip angle = 6˚, multi coil receiver coil, TE = 0.0035 s). For 3 participants, the clinical scan was an Axial T2 scan (slice thickness = 2mm, number of slices = 242, single-echo, echo train length = 141, TR = 2.8, flip angle = 90˚, multi coil receiver coil, TE = 0.48 s). Using tissue type–specific maps and individualised surface reconstructions, each electrode in grey matter was mapped to the nearest surface vertex, and labelled as the corresponding parcel, based on minimum GD from the centroid coordinate of the electrode to the cortical midsurface.

### Directed information processing

Intracranial EEG signals were re-referenced to the average signal of white matter channels to remove scalp reference and suppress far-field potentials caused predominantly by volume conduction [127]. The auto spectral density of each channel, $P_{xx}$, and the cross power spectral density between pairs of within-subject channels, $P_{xy}$, were calculated with Welch's method (59 overlapping blocks, 2-s duration, 1-s steps, weighted by Hamming window) [128]. These measures allow for the calculation of magnitude squared coherence between 2 signals [129, 130]:

$$C_{xy}(f) = P(f)_{xy}^2 / P(f)_{xx} P(f)_{yy}$$

In the above formula, $P_{xx}$ and $P_{yy}$ are power spectral density estimates, and $P_{xy}$ is the cross spectral density estimate. Coherence was evaluated in 0.77 Hz steps ($n = 129$) from 0.5 to 100 Hz. The 55 to 65 Hz range was not inspected due to power line noise at 60 Hz. We used boosting regression models to estimate variance explained in undirected coherence by the wiring space [see "Predicting functional connectivity" section for details]. In contrast to the fMRI-analysis, however, only within-sample variance explained ($R^2$) was examined.

The temporal coupling of 2 signals was determined by the phase slope index using the formula

$$\psi_{xy} = \Im \left( \sum_{f \in F} C_{xy}^*(f) C_{xy}(f + \delta f) \right)$$

In the above formula, $C_{xy}$ is complex coherence as defined above, $\delta f$ is the frequency resolution, $\Im(\cdot)$ denotes taking the imaginary part, and $F$ is the set of frequencies over which the slope is summed [49]. We used a sliding window approach for defining frequency bands, using 4-Hz bandwidth and 2-Hz overlap from 1 to 100 Hz, excluding the 55 to 65 Hz range due to power line artefact. The phase slope index leverages the relationship between increasing phase difference with increasing frequency to establish the driver and respondent sources [49]. Given incomplete coverage of intracranial electrodes in each participant, we discretised the wiring space into a set of subsections using consensus k-means clustering. Consensus-based k-means clustering and converged on a stable solution of k = 12 across 100 repetitions, provided

the k range of 10 to 20 [131]. All clusters were represented in the intracranial data (S12 Fig). We used a linear mixed-effect models to approximate the relationship between phase slope index within each frequency band and cluster membership:

$$\Psi \sim \text{categorical}(EdgeType) + \text{wd} + \Delta E1 + \Delta E2 + (1|\text{subject}) + (1|\text{channel}_{\text{seed}}) + (1|\text{channel}_{\text{target}}) + \varepsilon$$

Here, $\Psi$ stands for phase slope index, and *EdgeType* was defined by the seed and target clusters, in such that the *EdgeType* of a connection from cluster 1 to cluster 2 would be "12." *EdgeType* was a fixed effect, and *subject* and *channels* were nested in the model with random intercepts. We also included wiring distance (*wd*), difference on eigenvector 1 (*ΔE1*), and difference on eigenvector 2 (*ΔE2*) as fixed effects to account for variations in the positioning on nodes within a cluster. We used the t-statistics of each *EdgeType* category as a measure of phase slope index between clusters, then vectorised the t-statistics across all frequency bands for each cluster and used Pearson correlations to estimate the similarity of cluster's directed coherence patterns. For each cluster-to-cluster correlation, the phase slope index of that direct relationship was removed, thus the correlation indicates the similarity of phase slope index to all other clusters. A principle component analysis was used to extract the main axes of variation in the cluster similarity matrix. This component loading was cross-referenced with cell type–specific gene expression, with regions labelled by the corresponding cluster, as well as average externopyramidisation estimates for each cluster. Due to the limited number of observations to predictor variables, we opted for lasso regularisation and focused on high regularisation/sparsity models to characterise features importance [132]. The standard deviation in fitted least-squares regression coefficients was calculated using a leave-one-observation-out procedure, all of which used the range of lambdas from the full model. We performed a post hoc multiple linear regression with the sparsest model to evaluate the variance explained in the component loading by few cellular features (adjusted $R^2$). Next, we identified the most influential frequency bands to the component loading by performing Pearson correlations between the average phase slope index spectra with the component loading scores. Inspecting the frequency bands with maximum and minimum rho values, we performed significance thresholding of the cluster-to-cluster t-statistic matrices using the fixed effect *p*-values from the linear mixed-effect model with an alpha level of 0.05. Standard deviations in t-statistics were quantified via leave-one-subject-out iterations to ensure robustness of the direction and strength of the phase slope index estimates. Finally, we followed criteria from Felleman and van Essen [48] to test whether the frequency-specific phase slope index networks conformed to a hierarchical topology. We performed this in a "top-down" fashion by progressively adding clusters to lower levels of the model based on the driver–respondent relationships shown in the thresholded t-statistic matrices. First, clusters that only drive oscillations (i.e., positive t-statistics) were placed at the top level of the hierarchy, then the next level was populated by clusters that only respond to clusters in upper levels, and so forth. The internal consistency of the hierarchy is determined by whether all significant edges can be placed into the model with a constant flow of directed coherence from top to bottom. Preprocessing was performed using the Field-Trip toolbox [133], while the cross-spectral density estimates, phase slope index (http://doc.ml.tu-berlin.de/causality/), and linear models were estimated using MATLAB [134].

## Ethics statement

The study was approved by the local research ethics board of the Montreal Neurological Institute and Hospital (REB #2018–3469). Data were provided (in part) by the Human Connectome Project, WU-Minn Consortium (Principal Investigators: David Van Essen and Kamil

Ugurbil; 1U54MH091657) funded by the 16 NIH Institutes, and Centers that support the NIH Blueprint for Neuroscience Research, and by the McDonnell Center for Systems Neuroscience at Washington University.

## Supporting information

**S1 Fig. The multi-scale cortical wiring model with the Freesurfer-style preprocessing pipeline. (A)** Wiring features, i.e., GD, MPC, and diffusion-based TS were estimated between all pairs of nodes. **(B)** Normalised matrices were concatenated and transformed into an affinity matrix. Manifold learning identified a lower-dimensional space determined by cortical wiring. **(C)** Left: node positions in this newly discovered space, coloured according to proximity to axis limits. Closeness to the maximum of the second eigenvector is redness, towards the minimum of the first eigenvector is greenness, and towards the maximum of the first eigenvector is blueness. The first 2 eigenvectors are shown on the respective axes. Right: Equivalent cortical surface representation. **(D)** Calculation of interregional distances (isocontour lines) in the wiring space from specific seeds to other regions of cortex (left). Overall distance to all other nodes can also be quantified to index centrality of different regions, with more integrative areas having shorter distances to nodes (right). Essential data are available on https://git.io/JTg1l. a.u., arbitrary units; GD, geodesic distance; MPC, microstructure profile covariance; TS, tractography strength.
(TIF)

**S2 Fig.** (A) Scatterplots depict the global correlation between the different cortical wiring features, i.e., GD, MPC, and diffusion-based TS. Average r values for each node are projected below on the cortical surface. (B) Fingerprinting involved combining all 3 wiring features of 1 region. The SD across the 3 features was estimated for each edge, then the average was taken for each node as a measure of feature variance of the fingerprint. Essential data are available on https://git.io/JTg1l. GD, geodesic distance; MPC, microstructure profile covariance; SD, standard deviation; TS, tractography strength.
(TIF)

**S3 Fig. Independent replication of the wiring space.** The wiring space was regenerated using an independent MICs dataset of 40 healthy adults scanned at our imaging centre. Imaging parameters were similar to the original cohort, albeit using quantitative T1 relaxometry as a marker of cortical microstructure rather than HCP's T1w/T2w ratio mapping. **(A|C)** Scatterplots show marked correspondence between the original HCP "Discovery" sample eigenvectors and those from the MICs dataset, with Spearman correlations shown. **(B|D)** Using boosting regression, we similarly found that high, but regionally variable, variance in the group average functional connectivity could be explained by the wiring space in the MICs dataset. Essential data are available on https://git.io/JTg1l. HCP, Human Connectome Project; MICs, Microstructure Informed Connectomics.
(TIF)

**S4 Fig. Individual variation in the wiring space and prediction of rs-fMRI connectivity. (A)** Wiring spaces constructed in 2 individuals of the "Hold-out," which were aligned to the group-level "Discovery" wiring space. **(B)** MSE between predicted and empirical functional connectivity. Each box represents a node, and the distribution is taken across individuals. Box are coloured by the assignment of the node to a functional community (Fig 3). **(C–E)** Individual variation was calculated as the Euclidean distance between the node's position in the subject vs group average wiring space, where the position is synonymous with the values on the first 2 eigenvectors following Procrustes alignment. Interindividual variance was taken as

the average across all participants and provided as a.u. We also calculated the mean and SD in the MSE between predicted and empirical functional connectivity for each node across participants. Node-wise values are presented in the wiring space, on the cortical surface and stratified by level of laminar differentiation [7, 133] and functional network [49]. Essential data are available on https://git.io/JTg1l. a.u., arbitrary units; MSE, mean squared error; rs-fMRI, resting-state functional magnetic resonance imaging; SD, standard deviation.
(TIF)

**S5 Fig. Cytoarchitectural substrates of the wiring space with the Freesurfer-style preprocessing pipeline. (A)** A 3D postmortem histological reconstruction of a human brain [35] was used to estimate cytoarchitectural similarity and externopyramidisation. Here, we present a coronal slice, a drawing of cytoarchitecture, a magnified view of cortical layers in BigBrain, and a staining intensity profile with example of calculation of externopyramidisation [44]. **(B)** Matrix and density plot depict the correlation between BigBrain-derived cytoarchitectural similarity and wiring distance between pairs of regions. **(C)** Externopyramidisation, projected onto the cortical surface and into the wiring space, is highest at the bottom of the structural manifold. Essential data are available on https://git.io/JTg1l.
(TIF)

**S6 Fig. Transcriptomic substrates of the wiring space with the Freesurfer-style preprocessing pipeline. (A)** mRNA-seq probes, assigned to 11 representative nodes (coloured as in Fig 1C, i.e., their position in the wiring space), provided good coverage of the space, and enabled characterisation of cell type–specific gene expression patterns. Average cell type–specific gene expression patterns projected in the wiring space, with brighter colours signifying higher expression. **(B)** Equally spaced intercardinal axes superimposed on the wiring space, and below, line plots showing correlation of gene expression patterns with each of the axes. Colours correspond to the cell types shown in part A. **(C)** Strongest axis of variation (i.e., maximum |r|) in expression of each cell type overlaid on the structural manifold. Essential data are available on https://git.io/JTg1l. mRNA-seq, mRNA-sequencing.
(TIF)

**S7 Fig. Wiring space captures the interplay of more simple axes. (A)** Group average cortical thickness and curvature measures in the manifold. **(B)** Centroid coordinate of each parcel. x = left–right, y = posterior–anterior, z = inferior–superior. **(C|D)** Principles gradients from single modalities. Essential data are available on https://git.io/JTg1l.
(TIF)

**S8 Fig. From cortical wiring to functional connectivity with the Freesurfer-style preprocessing pipeline. (A)** Nodes in the wiring-derived coordinate system coloured by functional community [47], with the distribution of networks shown by density plots along the axes. **(B)** Wiring distance between nodes, ordered by functional community, revealed a modular architecture. **(C)** Violin plots show the average wiring distance for nodes in each functional community, with higher values being more specialised in their cortical wiring. **(D)** Using the boosting regression models from the "Discovery" dataset, we used features of the wiring space to predict z-standardised functional connectivity in a "Hold-out" sample. The model was enacted for each node separately. **(E)** MSE across nodes are shown in the wiring space and on the cortical surface (S3 Fig). **(F)** Predictive accuracy of various cortical wiring models, involving the use of different features, multifeature fusion, eigenvectors from diffusion map embedding, and a linear or ML learner. **(G)** MSE of the wiring space model stratified by functional community. Essential data are available on https://git.io/JTg1l. ΔE1, difference on eigenvector 1; ΔE2, difference on eigenvector 2; FC, functional connectivity; ML, machine learning; MSE,

mean squared error; WD, wiring distance.
(TIF)

**S9 Fig. Modelling fMRI-derived connectivity from the wiring space.** **(A)** Box and spaghetti plots depict the relative importance of wiring space features for the boosting regression model at each node. **(B)** Learning rate and number of estimated selected by cross-validation for the model at each node. Essential data are available on https://git.io/JTg1l. fMRI, functional magnetic resonance imaging.
(TIF)

**S10 Fig. Patient wiring space.** The wiring space generated from the group average of cortical wiring features across the patients was highly similar to the healthy template (correlations between eigenvectors 1/2: r = 0.92/0.89). Essential data are available on https://git.io/JTg1l.
(TIF)

**S11 Fig. Large-scale organisation of coherence in the structural manifold with the Freesurfer-style preprocessing pipeline.** **(A)** Intracerebral implantations of 10 epileptic patients were mapped to the cortical surface and intracortical EEG contacts selected. We studied 5 minutes of wakeful rest. **(B)** Mean and SD in the variance explained in undirected coherence by wiring space features using adaboost machine learning across all nodes. **(C)** Clusters of the wiring space. **(D)** Phase slope index ($\Psi$) was calculated for each pair of intra-subject electrodes, then cluster-to-cluster estimates were derived from a linear mixed-effect model. Pearson correlation across $\Psi$ estimates was used to measure the similarity of clusters, and the major axis of regional variation was identified via principle component analysis. (i) Average $\Psi$ spectra for each region coloured by loading on the first principle component (accounting for 39% of variance). (ii) Component loadings presented in the wiring space and on the cortical surface illustrate a gradient from upper left regions, corresponding to central areas, towards lower right areas, corresponding to temporal and limbic areas. (iii) Lasso regularisation demonstrates the contribution of cell type–specific gene expression (colours matching Fig 3) and externopyramidisation (red) to explain variance in the component loadings. Shaded areas show the SD in fitted least-squares regression coefficients across leave-one-observation-out iterations. For example, inhibitory neurons expression levels (green) are closely related to the component loading, as shown in the scatterplot. Essential data are available on https://git.io/JTg1l. EEG, electroencephalography; SD, standard deviation.
(TIF)

**S12 Fig. Robustness of intracranial EEG analyses.** **(A)** Matrix depicting the number of intra-subject electrode pairs that contribute to the cluster-to-cluster estimations of phase slope index. **(B)** Variance explained in coherence by boosting regression, operationalised at a single subject level with individualised wiring spaces. **(C)** Variance in coefficients from a principle component analysis of intercluster similarities in the phase slope index using a leave-one-subject-out procedure. **(D)** and **(E)** involve the same leave-one-subject-out procedure to gauge the variance in edge-wise phase slope index estimates at frequencies of interest. Notably, the range of estimates does not pass 0 for significant edges. Essential data are available on https://git.io/JTg1l. EEG, electroencephalography.
(TIF)

**S13 Fig. Hierarchical information processing is organised within the structural manifold with the Freesurfer-style preprocessing pipeline.** "Top": The most influential frequencies on the component loading were identified through a correlation of average $\Psi$ spectra (**Fig 5Di**) with the component loading. For the global maxima (28 Hz, left) and minima (97 Hz, right),

we present cluster-to-cluster $\Psi$ estimates, thresholded at $p < 0.05$. "Bottom": Suprathreshold edges are plotted as a hierarchical schema, where the directed coherence estimates indicate flow of oscillations from the top to the bottom of the hierarchy. The hierarchical level of each cluster is also presented on the cortical surface and in the wiring space, illustrating unique spatial patterns of the frequency-specific hierarchies. Compared to the HCP-style pipeline, the frequencies of interest are slightly shifted resulting in distinct hierarchies. Essential data are available on https://git.io/JTg1l. HCP, Human Connectome Project.
(TIF)

**S1 Text. Supplementary Tables A–C.**
(DOCX)

# Acknowledgments

The authors would also like to express their gratitude to the open science initiatives that made this work possible, including the teams involved in the BigBrain project, the PsychEncode consortium, the Human Connectome Project, and Scikit-learn. Furthermore, we thank Dr Matthias Kirschner for his helpful insight on the manuscript and the MRI and EEG technicians at the Montreal Neurological Institute.

# Author Contributions

**Conceptualization:** Casey Paquola, Jakob Seidlitz, Petr Klimes, Richard A. I. Bethlehem, Reinder Vos de Wael, Jonathan Smallwood, Boris C. Bernhardt.

**Data curation:** Casey Paquola, Jessica Royer, Sara Larivière, Raul Rodríguez-Cruces, Boris C. Bernhardt.

**Formal analysis:** Casey Paquola, Oualid Benkarim, Petr Klimes, Sara Larivière, Reinder Vos de Wael.

**Funding acquisition:** Boris C. Bernhardt.

**Investigation:** Casey Paquola, Jeffery A. Hall, Birgit Frauscher.

**Methodology:** Casey Paquola.

**Project administration:** Boris C. Bernhardt.

**Resources:** Raul Rodríguez-Cruces.

**Software:** Casey Paquola.

**Supervision:** Boris C. Bernhardt.

**Visualization:** Casey Paquola.

**Writing – original draft:** Casey Paquola, Jonathan Smallwood, Boris C. Bernhardt.

**Writing – review & editing:** Casey Paquola, Jakob Seidlitz, Oualid Benkarim, Jessica Royer, Petr Klimes, Richard A. I. Bethlehem, Sara Larivière, Reinder Vos de Wael, Birgit Frauscher, Jonathan Smallwood, Boris C. Bernhardt.

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
