## [Editor Report · Decision Letter 0]

10 Apr 2020

Dear Dr Bernhardt, 

Thank you for submitting your manuscript entitled "A multi-scale cortical wiring space links cellular architecture and functional dynamics in the human brain" for consideration as a Research Article by PLOS Biology.

Your manuscript has now been evaluated by the PLOS Biology editorial staff, as well as by an academic editor with relevant expertise, and I'm writing to let you know that we would like to send your submission out for external peer review.

Please re-submit your manuscript within two working days, i.e. by Apr 15 2020 11:59PM.

Kind regards,

Roli Roberts

Senior Editor

PLOS Biology

---

## [Decision Letter · Decision Letter 1]

4 Jun 2020

Dear Dr Bernhardt,

Thank you very much for submitting your manuscript "A multi-scale cortical wiring space links cellular architecture and functional dynamics in the human brain" for consideration as a Research Article at PLOS Biology. Your manuscript has been evaluated by the PLOS Biology editors, an Academic Editor with relevant expertise, and in this case by two independent reviewers. We had recruited a third reviewer, but they have not been able to submit in a timely fashion; if they send us any comments we shall forward them to you.

You'll see that both reviewers are broadly positive about you study, but each raises some concerns that must be addressed before further consideration. While reviewer #1 only has some modest requests, reviewer #2 suggests a number of improvements to the methods (metrics of distance, parcellation, treatment of task vs resting-state data, etc.) and wonders whether your presentation might benefit from breaking up the Figures somewhat (we would tend to agree with the reviewers' comments on this point).

In light of the reviews (below), we will not be able to accept the current version of the manuscript, but we would welcome re-submission of a much-revised version that takes into account the reviewers' comments. We cannot make any decision about publication until we have seen the revised manuscript and your response to the reviewers' comments. Your revised manuscript is also likely to be sent for further evaluation by the reviewers.

We expect to receive your revised manuscript within 2 months. 

**IMPORTANT - SUBMITTING YOUR REVISION**

*Re-submission Checklist*

*Published Peer Review*

*PLOS Data Policy*

*Blot and Gel Data Policy*

Sincerely,

Roli Roberts

Senior Editor

PLOS Biology

REVIEWERS' COMMENTS:

Reviewer #1:

Thank you for inviting me to review this manuscript by Paquola and colleagues, in which the authors conducted a sophisticated multi-modal analysis of human brain imaging data in an effort to better characterize the relationship between structure and function. The method leverages recent results from this group that have linked microstructural heterogeneity with macroscale organization. The results of this study are compelling, and extend their previous work into new domains, including more detailed micro-structural properties (from the BigBrain atlas) and high temporal resolution structural EEG data. The methods are well-motivated and described, and the statistical approaches are all appropriately justified.

I have a few minor points that I hope will help to improve the manuscript:

Did the authors consider any other dimensionality reduction approaches? For instance, principal component analysis is similar to diffusion map embeddings, and can provide even more robust results than DME when applied to data that are well-suited to linear dimensionality reduction.

I'm not sure that I fully appreciate the authors interpretation of the 'integrative core'. From my understanding, regions that have small loadings onto the wiring space (i.e., those that exist in the middle of the plot in Figure 1D) are those that explained the least variance (at least with respect to the first two eigenvectors). It's not clear that identifying those particular regions has a specific interpretation per se. It's also not really clear precisely how the authors identified the 'integrative core'. I recommend that the authors expand on this section, and highlight precisely how the metrics were calculated in the main text.

Reviewer #2:

[identifies himself as Matthew F. Glasser M.D. Ph.D.]

The authors present an interesting multimodal manuscript investigating a core wiring coordinate system in the human brain. A strength of this paper is the martialing of a very diverse and comprehensive set of modalities to compare with their model. The paper is let down by some puzzling methodological decisions. Also, it is a bit challenged by the very dense figures, which may be hard for some readers to parse, though I recognize that the authors are trying to cover a lot of material. A table summarizing the MSE and R2 values for various analyses might also be beneficial. 

I have the following major concerns about the methods:

1) I didn't understand the rationale for the modified geodesic distance measure. If the purpose is to measure distance across the cortical sheet, the standard approach to geodesic distance is more appropriate. If the purpose is to measure putative axonal length, a better approach would be to actually measure distance using tractography (which can be done with FSL's probtrackx2 tool). Given the critical importance of distance to the overall measure, it seems that the distance-based approach must be more clearly neurobiologically motivated.

2) Why not use the HCP's multi-modal cortical parcellation from Glasser et al., 2016 Nature or some other functionally relevant parcellation? A random parcellation based on gyral and sulcal boundaries is unlikely to be optimal for either neurobiological interpretation or computation of functional connectivity. I made a similar comment in the author's prior paper on myelin covariance. 

3) The task fMRI data are used for functional connectivity, but it is not said if they were denoised or not. Also, it is unclear why the connectivity measure is changed for task functional connectivity. Task data cannot be "made into" resting state data, so it would be better to just use the same methods that one uses with resting state data. There are both similarities and differences between the functional networks that make up HCP resting state and task fMRI data as shown in Glasser et al., 2018 Neuroimage. 

Minor Concerns:

1) It is noted that HCP subjects were asked not to fall asleep, but it is known that many did (Glasser et al., 2018 Neuroimage). Not sure if this affects the current study though. 

2) The HCP resting state and task fMRI scan length are incorrect: Resting state was a total of 3456 seconds across 4 runs and task was 2793.6 seconds across 14 runs. 

3) "then subjected to ICA-FIX for removal of additional noise" sounds kind of pejorative. Please rephrase. 

4) The immediately following sentence should be reworded: "The data were resampled from volume to MSMAll functionally aligned surface space."

5) I still don't understand the oversampling of the surface profiles. Even for 0.7mm isotropic HCP data, for a thick area at 4mm, this means an average of 0.33mm between surfaces and for a thin area at 1.6mm an average of 0.13mm between surfaces. 

6) Please clarify that the equal volume surfaces used are from a similar implementation to that in Connectome Workbench's -surface-cortex-layer command or Waehnert et al., 2014 Neuroimage. 

7) I didn't follow the "log-domain intensity normalization" of "fiber orientation distributions."

8) What is meant by "spike regression" in the resting state denoising. This is not a step recommended by the HCP for their data.

9) What approach was used for surface registration in the replication data? Folding or functional? Using FreeSurfer or MSM?

---

## [Decision Letter · Decision Letter 2]

3 Aug 2020

Dear Dr Bernhardt,

Thank you very much for submitting a revised version of your manuscript "A multi-scale cortical wiring space links cellular architecture and functional dynamics in the human brain" for consideration as a Research Article at PLOS Biology. This revised version of your manuscript has been evaluated by the PLOS Biology editors, the Academic Editor and one of the original reviewers.

You'll see that reviewer #2 continues to have some concerns that will need to be addressed. The Academic Editor has asked me to emphasise to you that we agree with this reviewer on the three major points raised, and indeed, for clarity's sake, I have included some of the comments from the Academic Editor at the foot of this email.

I should say that we are prepared to consult the reviewer one last time on this manuscript, and if you fail to satisfy them then, we may not consider your manuscript further.

In light of the reviews (below), we will not be able to accept the current version of the manuscript, but we would welcome re-submission of a much-revised version that takes into account the reviewers' comments. We cannot make any decision about publication until we have seen the revised manuscript and your response to the reviewers' comments. Your revised manuscript is also likely to be sent for further evaluation by the reviewers.

We expect to receive your revised manuscript within 3 months. 

**IMPORTANT - SUBMITTING YOUR REVISION**

*Re-submission Checklist*

*Published Peer Review*

*PLOS Data Policy*

*Blot and Gel Data Policy*

Sincerely,

Roli Roberts

Senior Editor,

rroberts@plos.org,

PLOS Biology

REVIEWERS' COMMENTS:

Reviewer #2: 

[identifies himself as Matthew F. Glasser]

The authors have incompletely addressed my concerns. I continue to have the following major concerns, and suggest paths to resolving them that hopefully will be acceptable to the authors:

1) This statement in the response to reviews: "Denoising of task fMRI was performed within the HCP minimal preprocessing pipeline (Glasser et al.2013, Neuroimage), including motion correction, EPI distortion correction and exclusion of voxels with locally high coefficient of variation in the volume-to-surface mapping." really concerns me. The HCP's minimal preprocessing pipelines are designed to remove spatial distortions, realign the data within and across modalities, and to exclude voxels that are likely to contain little usable signal due to corruption from blood vessels or dura. They do not do ANY temporal denoising. The HCP's recommended approach to temporal denoising includes at a minimum spatial ICA-based cleanup (such as that provided by the sICA+FIX pipeline that was run on the resting state data). Unfortunately, such cleanup has not yet been released for the HCP's task fMRI data (but will be in the future); however, it is absolutely critical for a connectivity-based analysis of fMRI data. When looking at properly cleaned HCP data (both resting state and task) there is no increase in outliers in task data over resting state (indeed the kurtosis of resting state data is slightly higher than task data). Thus, the authors are likely misattributing the effects of strong structured artifacts that have not been removed from the task fMRI data to the task condition itself, which is inaccurate. I don't think the task analysis is critical to the paper, so if it cannot be easily done correctly, perhaps it is best removed/saved for when properly denoised task fMRI data is available. 

2) I appreciate the authors including the analyses from the HCP parcellation and Schaefer parcellation. They do seem both to be quite a bit different from the gyral and sulcal parcellation results and more similar to each other. This is expected, given that both have significant contributions from resting state functional connectivity, which has much more neurobiological relevance than folding patterns. Thus, it seems the main analyses of the paper should be based around one or both of these and not the outlier folding parcellation. It is also not clear to me what neurobiological property of brain organization the authors are referring to when they say "structural neuroanatomy." It isn't one of the 4 cortical properties that have been used to define cortical areas either classically or in modern maps (i.e. cortical microstructural architecture, cortical function as measured invasively or non-invasively, connectivity—functional, diffusion based, or invasive tracer based, or topographic maps). 

3) The question of the correct distance metric is still unclear to me. I think the authors are intending their distance metric to represent "distance along an axon connecting two brain areas" and not "distance along the surface." Their inclusion of pure geodesic distance does illustrate that geodesic distances are largely driving the current results. Although the authors are correct that the optimal approach to deriving tractography connection weights has not been found, tractography algorithms are reasonably good at measuring the physical distance between grey matter points along the most probable connection path (e.g. Donahue et al 2016 Journal of Neuroscience). Without experimental evidence, I don't have confidence the authors' analyses would not be meaningfully different if they used estimated axonal distances. If they don't wish to do these analyses, they need to better justify why along axonal distances are not the correct measure for their analysis of brain "wiring." 

Minor Comment:

1) Describe the replication analysis surface registration as "Timeseries were sampled on native cortical surfaces and resampled to fsaverage via folding-based FreeSurfer surface registration."

COMMENTS FROM THE ACADEMIC EDITOR (lightly edited):

I think that in relation to [reviewer #2's] point 2 the authors have already included the approach recommended by R2 but they have not removed their old approach. The authors are arguing that their approach produces similar results but R2 is saying that this is not really the case. Here I would strongly argue that the authors should follow R2’s advice.

I think that in relation to point 1, R2 has given a clear suggestion for how to deal with an fMRI pre-processing problem or he has suggested simply removing this aspect of the results. Again I am inclined to agree with R2 that this pre-processing step is needed. R2 was instrumental in creating the Human Connectome Project (HCP) and I think that he is, in general, well informed about it. I think that, here again, there is a clear way forward for the authors to take (or perhaps two alternative routes that might be taken).

Major point 3 – here it is very difficult to understand the authors’ reasoning behind their axon length estimates and it seems that R2 is correct to question them.

---

## [Editor Report · Decision Letter 3]

16 Oct 2020

Dear Dr Bernhardt,

Thank you for submitting your revised Research Article entitled "A multi-scale cortical wiring space links cellular architecture and functional dynamics in the human brain" for publication in PLOS Biology. The Academic Editor has now checked your revisions and responses to reviewer #2. 

Based on the Academic Editor's assessment, we're delighted to let you know that we're now editorially satisfied with your manuscript. However before we can formally accept your paper and consider it "in press", we also need to ensure that your article conforms to our guidelines. A member of our team will be in touch shortly with a set of requests. As we can't proceed until these requirements are met, your swift response will help prevent delays to publication.

IMPORTANT: Please also make sure to address the data and other policy-related requests noted at the end of this email.

- a cover letter that should detail your responses to any editorial requests, if applicable

*Copyediting*

*Published Peer Review History*

*Early Version*

Sincerely,

Roli Roberts

Senior Editor,

rroberts@plos.org,

PLOS Biology

ETHICS STATEMENT:

-- Please include information about the form of consent (written/oral) given for research involving human participants. All research involving human participants must have been approved by the authors' Institutional Review Board (IRB) or an equivalent committee, and all clinical investigation must have been conducted according to the principles expressed in the Declaration of Helsinki. We note that you say "The protocol received prior approval from the MNI Institutional Review Board.” Please could you supply the protocol approval number for this?

DATA POLICY:

Regardless of the method selected, please ensure that you provide the individual numerical values that underlie the summary data displayed in all figure panels as they are essential for readers to assess your analysis and to reproduce it. NOTE: the numerical data provided should include all replicates AND the way in which the plotted mean and errors were derived (it should not present only the mean/average values).

IMPORTANT: We note that your Data Availability Statement mentions the HCP, and also two Github depositions that involve intermediate data and scripts. These depositions look very comprehensive, but it remains very unclear how they relate to the current Figures, especially given they haven’t been updated since the first submission. Please clarify this and/or provide the data that directly underlie all of the main and supplementary Figures. Please also ensure that figure legends in your manuscript include information on where the underlying data can be found, and ensure your supplemental data file/s has a legend.

---

## [Editor Report · Decision Letter 4]

2 Nov 2020

Dear Dr Bernhardt,

On behalf of my colleagues and the Academic Editor, Matthew F. S. Rushworth, I am pleased to inform you that we will be delighted to publish your Research Article in PLOS Biology. 

PRODUCTION PROCESS

Before publication you will see the copyedited word document (within 5 business days) and a PDF proof shortly after that. The copyeditor will be in touch shortly before sending you the copyedited Word document. We will make some revisions at copyediting stage to conform to our general style, and for clarification. When you receive this version you should check and revise it very carefully, including figures, tables, references, and supporting information, because corrections at the next stage (proofs) will be strictly limited to (1) errors in author names or affiliations, (2) errors of scientific fact that would cause misunderstandings to readers, and (3) printer's (introduced) errors. Please return the copyedited file within 2 business days in order to ensure timely delivery of the PDF proof. 

If you are likely to be away when either this document or the proof is sent, please ensure we have contact information of a second person, as we will need you to respond quickly at each point. Given the disruptions resulting from the ongoing COVID-19 pandemic, there may be delays in the production process. We apologise in advance for any inconvenience caused and will do our best to minimize impact as far as possible.

EARLY VERSION

PRESS 

Kind regards,

Alice Musson

Publishing Editor, 

PLOS Biology

on behalf of

Roland Roberts,

Senior Editor

PLOS Biology